# PTPN1/PTPN2 inhibition improves NK cancer therapy by enhancing IL-2 and mitigating TGFβ1 responses

Chu-Han Feng [1,2], Linda Peltier [3,4], Tiffanie Chouleur[1,5,6], Milea DiPonzio [1,2], Isabelle Aubry[1], Alexandre J Poirier[1,5], Zuzet M Cordova[1], Yunyun Shen[1,7], Sébastien Tabariès [1], Xiaona Cao[1,8], Guojun Chen[1,8], Andreas Bikfalvi [6], Silvia M Vidal[9,10], Peter M Siegel [1,5,7,11], Pierre Laneuville[3,4,5] & Michel L Tremblay [1,2,5,7 ✉]

## Abstract

**Natural killer (NK) cells are promising candidates for allogeneic anti-cancer immunotherapy. However, their cytolytic activity is often suppressed by the tumor microenvironment. We demonstrate that genetic silencing or pharmacological dual inhibition of protein tyrosine phosphatases PTPN1 and PTPN2 (PTPN1/N2) in NK cells significantly enhances anti-tumor cytolytic activity both in vitro and in vivo. This augmented NK cell activity is mediated by increased expression of early activation markers and the production of effector molecules such as granzyme B and interferon-gamma (IFN-γ). Notably, this elevated cell cytolytic response remains substantially resistant to the immunosuppressive effects of TGFβ-1, a cytokine known to dampen NK cell activity and commonly present in the tumor microenvironment. Mechanistically, targeting PTPN1/N2 in NK cells promotes JAK/STAT signaling pathways and sensitizes cells to IL-2 stimulation. Importantly, dual inhibition of PTPN1/N2 markedly enhances the cytolytic activity of cord blood NK cells against patient-derived glioblastoma cells, highlighting the potential of this approach for future therapeutic applications. These findings provide compelling evidence that dual targeting of PTPN1/N2 could significantly improve the efficacy of therapeutic "off-the-shelf" NK cell-based immunotherapy.**

**Keywords** Anti-cancer Immunotherapy; Cytokine Signaling; NK Cells; PTPN1 (PTP1B); PTPN2 (TC-PTP)
**Subject Categories** Cancer; Immunology; Signal Transduction

## Introduction

Natural killer (NK) cells are the first-line responders to recognize and eliminate malignantly transformed or pathogenically infected cells without prior antigen sensitization (Herberman et al, 1975; Lanier, 2024; Vivier et al, 2024). NK cells are competitive candidates to generate *off-the-shelf* adoptive cell therapy, and they can be obtained from allogeneic sources such as peripheral blood or umbilical cord blood (CB) where the NK cell population is enriched (Chen et al, 2024; Laskowski et al, 2022; Theilgaard-Monch et al, 2001). Clinical evidence in hematologic cancer NK therapy indicates a low risk of developing graft-versus-host diseases (GvHD) and severe systemic inflammation, possibly due to a mix of pro-inflammatory (e.g., IFN-γ and TNF-α) and anti-inflammatory (e.g., IL-10 and TGFβ1) cytokines secreted by activated NK cells to modulate immune responses(Cooper et al, 2001; Laskowski et al, 2022; Ruggeri et al, 2002; Yu et al, 2006). Conversely, cytokines within the tissue microenvironment influence NK cell activation, maturation, and survival (Abel et al, 2018; Brady et al, 2010). Previous studies with murine and human NK cells showed that type I interferons and interleukins (IL-2, IL-12, IL-15, IL-18, and IL-21) augmented cytolytic responses, while immunosuppressive cytokines (TGFβ1, IL-35) dampen cytolytic functions (Vivier et al, 2024; Wu et al, 2017; Zwirner and Domaica, 2010). Given that NK cells within the tumor microenvironment receive a complex combination of cytokine signals from both local and distant sources, understanding how NK cells integrate mixed cytokine stimulations is crucial to improving the current NK cell-based anti-tumor therapy designs.

Homologous non-receptor protein tyrosine phosphatases type 1 (or PTP1B, encoded by gene *PTPN1*) and type 2 (or TC-PTP, encoded by gene *PTPN2*) regulate adaptive and innate immune cell

[1]Rosalind and Morris Goodman Cancer Institute, School of Biomedical Sciences, Faculty of Medicine, McGill University, Montreal, QC H3A 1A3, Canada. [2]Department of Microbiology and Immunology, School of Biomedical Sciences, Faculty of Medicine and Health Sciences, McGill University, Montreal, QC H3A 2B4, Canada. [3]Cellular Therapy Laboratory, Research Institute - McGill University Health Centre, Montreal, QC H4A 3J1, Canada. [4]Division of Hematology, McGill University Health Centre, Montreal, QC H4A 3J1, Canada. [5]Division of Experimental Medicine, Department of Medicine, Faculty of Medicine and Health Sciences, McGill University, Montreal, QC H4A 3J1, Canada. [6]Université de Bordeaux, INSERM, U1312 BRIC (Bordeaux Institute of Oncology), 33615 Pessac, France. [7]Department of Biochemistry, School of Biomedical Sciences, Faculty of Medicine and Health Sciences, McGill University, Montreal, QC H3G 1Y6, Canada. [8]Department of Biomedical Engineering, School of Biomedical Sciences, Faculty of Medicine and Health Sciences, McGill University, Montreal, QC H3A 2B4, Canada. [9]Department of Human Genetics, School of Biomedical Sciences, Faculty of Medicine and Health Sciences, McGill University, Montreal, QC H3A 1Y2, Canada. [10]Dahdaleh Institute of Genomic Medicine, McGill University, Montreal, QC H3A 0G1, Canada. [11]Department of Anatomy and Cell Biology, Faculty of Medicine, McGill University, Montreal, QC H3A 0C7, Canada. ✉E-mail: michel.tremblay@mcgill.ca

development and functions (Simoncic et al, 2006). In cancer cells, deficiency in PTPN1 and/or PTPN2 improved sensitivity to anti-PD-1 therapy and IFN-γ-mediated growth inhibition (Baumgartner et al, 2023; Manguso et al, 2017). PTPN1 and PTPN2 share structural and enzymatic similarities, possessing a secondary substrate-binding pocket that recognizes tandem phosphor-tyrosine (pTyr) sites and permits their simultaneous inhibition (Barford et al, 1994; Barr et al, 2009; Iversen et al, 2002). Despite their similarities, in vitro and in vivo studies suggested non-redundant roles for PTPN1 and PTPN2 in cell signaling and physiological response. These two PTPs have overlapping and specific substrate binding preferences, which may be partially due to their subcellular localization (Doody et al, 2009; Dube and Tremblay, 2004; Myers et al, 2001; Simoncic et al, 2002; Tiganis et al, 1998). Functional disparities are evident in whole-body knockout mice, where *Ptpn2* null mice die at 3–5 weeks of severe anemia and progressive systemic inflammatory diseases (You-Ten et al, 1997), while *Ptpn1* null mice exhibit an expected average life span and resistance to diet-induced obesity and diabetes (Elchebly et al, 1999; Klaman et al, 2000). Previous research indicates context-dependent and non-overlapping roles for both enzymes in regulating IFN-γ, IL-12, leptin, insulin receptor signaling, and endoplasmic reticulum (ER) stress responses (Dodd et al, 2019; Heinonen et al, 2009; Penafuerte et al, 2017; Tiganis, 2013).

PTPN1 and PTPN2 act as negative regulators of cytokine receptor signaling in T cells, B cells, macrophages, and dendritic cells, targeting JAK and STAT family members as substrates (Heinonen et al, 2009; Penafuerte et al, 2017; Pike and Tremblay, 2016). Protein-protein interaction assays by substrate trapping revealed substrate preferences, with JAK2 and TYK2 being PTPN1's preferred substrates in response to IFN-α or IFN-γ stimulations, and JAK1 and JAK3 being PTPN2's preferred substrates in response to IL-2 stimulation (Myers et al, 2001; Simoncic et al, 2002). Changes in PTPN1 and PTPN2 expression levels influence substrate binding preferences among STAT family members. Complete deficiencies of PTPN1 or PTPN2 promoted phospho-STAT5 and phospho-STAT3 signals, while partial reduc-tion of their enzymatic activities increased phospho-STAT1 and phospho-STAT4 signals in murine dendritic cells (Penafuerte et al, 2017). Given the importance of JAK/STAT signaling pathways in NK cells, we aimed to understand how these two enzymes regulate cytokine-activated NK cell cytolytic functions.

Herein, we report that pharmacological dual inhibition of PTPN1 and PTPN2 (PTPN1/N2) enhanced cytolytic functions in NK-92, an FDA-approved therapeutic human NK cell line, and CB NK cells against a broad range of tumors in vitro, including chronic and acute myeloid leukemia, glioblastoma, kidney cancer and metastatic breast cancer cell lines. Mechanistically, the increased cytolytic response is associated with increased production of IFN-γ, perforin, and granzyme B. Genetic silencing or pharmacological targeting of PTPN1/N2 in NK-92 cells upregu-lated activation markers CD69 and CD25, thereby rendering cells more sensitive to IL-2-induced JAK/STAT signaling. Furthermore, even under suppression by TGFβ1, PTPN1/N2-dually inhibited NK-92 cells retain higher anti-tumor cytolytic activity. In U87 glioblastoma and MDA-MB-231TR xenografted mouse models, PTPN1/N2 dual inhibition in NK-92 cells effectively delays tumor growth. Additionally, PTPN1/N2 dual inhibition enhances CB NK cell cytolytic functions against patient-derived

glioblastoma cells. Altogether, our results suggest that PTPN1 and PTPN2 are pharmacologically targetable for dual inhibition, with high potential to improve NK cell-based immunotherapies even in the context of an immunosuppressive tumor microenvironment.

# Results

## PTPN1 and PTPN2 negatively regulate human NK cells activation

While SHP-1 (*PTPN6*) and SHP-2 (*PTPN11*) are recognized as important regulators of NK cell activation (Olcese et al, 1996; Purdy and Campbell, 2009; Yusa et al, 2002), interestingly, *PTPN1* mRNA expression levels are much higher compared to those in NK-92 cell line. In addition, *PTPN1* mRNA was more abundant than *PTPN2* in human NK cell line NK-92 and cord blood NK cells (Fig. 1A,B).

To examine PTPN1 and PTPN2 functions in NK cells, we generated lentivirus-mediated shRNA knockdown (KD) NK-92 cells targeting *PTPN1* (sh*PTPN1*), *PTPN2* (sh*PTPN2*), or both PTPs (sh*PTPN1*/sh*PTPN2* double KD; denoted as dKD) (Figs. 1C and E-V1A,B). Genetic silencing of *PTPN1* or *PTPN2* alone or together did not affect NK cell viability under standard cell culture conditions (Fig. EV1C). However, PTPN1 KD, alone or with PTPN2 in dKD cells, were increased in cell size from 164 μm² (sh*FF*, $n = 707$) to 182 μm² (sh*PTPN1*, $n = 700$) and 183.5 μm² (dKD, $n = 742$) (Fig. EV1D–F). Since it has been known that NK cells increase in size after activation (Ng et al, 2019), we then examined the activation markers CD25 and CD69 on these NK cell lines with or without tumor target K-562, a chronic myelogenous leukemia cell line, stimulations. Our results indeed revealed that both the mean fluorescent intensity (MFI) of CD25 and the proportion of CD25-positive cells were significantly increased in sh*PTPN1* and sh*PTPN2* cells compared to control sh*FF* cells. Notably, dKD cells exhibited sub-additive effects on CD25 expression (Figs. 1D and EV1G,H). In contrast, while genetic silencing of *PTPN2* alone did not significantly alter CD69 expression levels, *PTPN1* knockdown induced higher CD69 expression and increased frequency of CD69-positive cells. Similar to CD25, dKD cells displayed synergistic effects of dual silencing *PTPN1* and *PTPN2*, further enhancing CD69 expression and the proportion of positive cells, both with and without stimulation (Figs. 1E and EV1I,J).

## Dual pharmacological inhibition of PTPN1 and PTPN2 enhanced anti-tumor cytolytic functions

The above results strongly suggest that inhibition of PTPN1 and PTPN2 enhances NK cell cytolytic function. Therefore, we next performed NK cell cytolytic assays. Genetic silencing of *PTPN1*, *PTPN2*, or both significantly improved direct cytolysis against NK-sensitive tumor K-562 cells across various effector-to-target-cell ratios (Figs. 2A and EV2A). Sh*PTPN1* and dKD cells outperformed sh*PTPN2* cells in augmenting NK-92 cell cytolysis (Figs. 2A and EV2A). Since the dual knockdown of PTPN1 and PTPN2 activates NK-92 cells to the highest level, we subsequently focused on examining the effects of pharmacological dual inhibition of PTPN1 and PTPN2 in human NK cells.

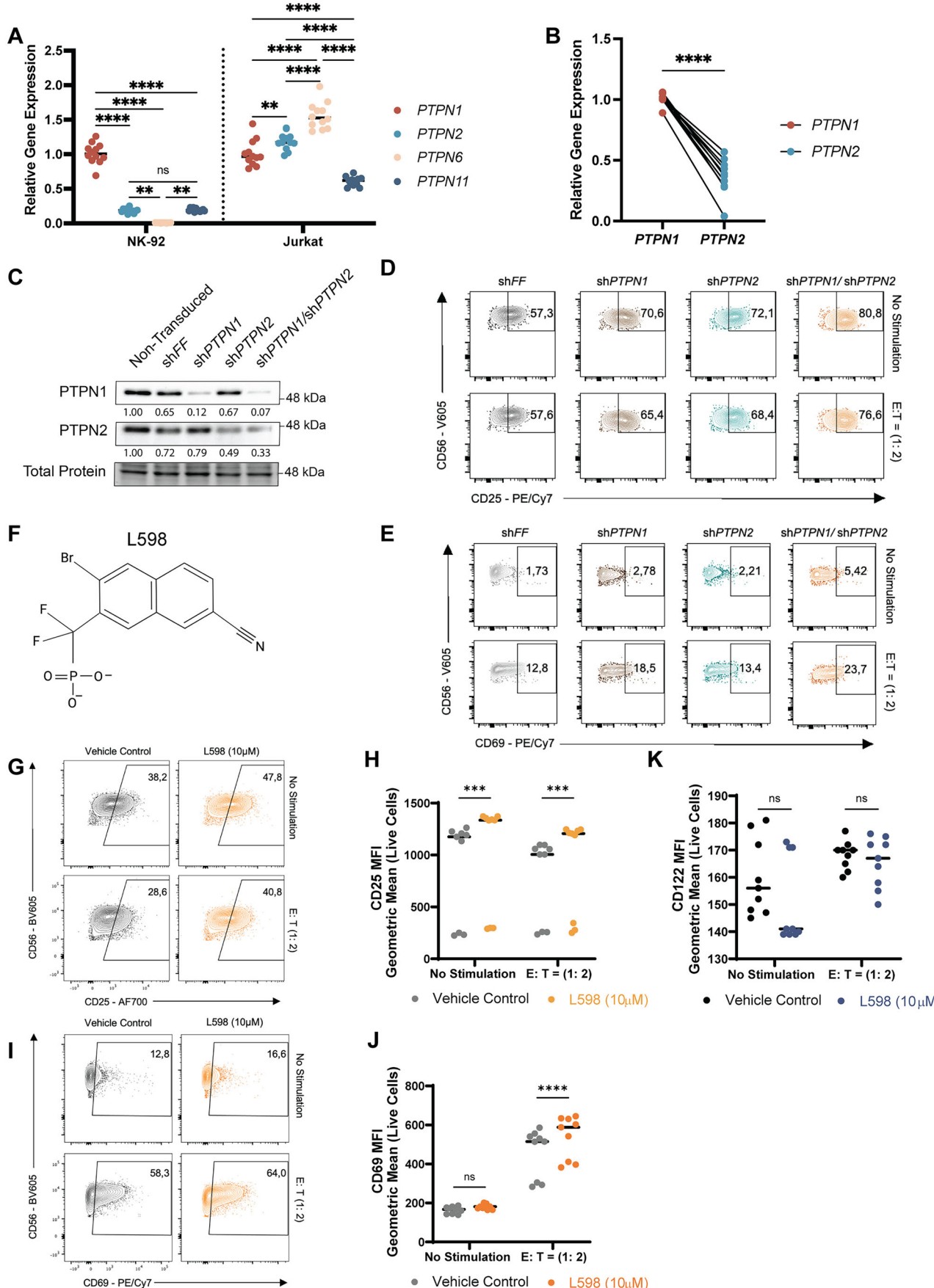

◀ **Figure 1. Deficiency of PTPN1 and PTPN2 upregulates activation markers CD25 and CD69.**

(A) qRT-PCR analysis of *PTPN1*, *PTPN2*, *PTPN6*, and *PTPN11* in human NK-92 cell line and CD4$^+$ T cell line Jurkat, with relative expression levels of all four PTP genes compared against *PTPN1*. Experiments were from $n$ = three biological replicates, each tested in four technical replicates. *P*-values: NK-92 cells: $p > 0.9999$ (ns) for *PTPN2* vs. *PTPN11*, $p = 0.0029$ (**) for *PTPN2* vs. *PTPN6*, $p = 0.0016$ (**) for *PTPN6* vs. *PTPN11*, $p \leq 0.0001$ (****); Jurkat cells: $p = 0.0042$ (**) for *PTPN1* vs. *PTPN2*, $p \leq 0.0001$ (****) (Two-way ANOVA with Šídák's multiple comparisons test). (B) qRT-PCR analysis of *PTPN1* and *PTPN2* in cord blood (CB) NK cells ($n$ = 10 donors each in quadruplicates), with *PTPN2* expression levels compared relative to *PTPN1*. *P*-value: $p \leq 0.0001$ (****) (Paired t-test). (C) Western blot analysis of PTPN1 and/or PTPN2 knockdown in shRNA-modified NK-92 cells, shown in a representative blot from $n$ = three biological replicates. (D, E) Fow cytometry blots of CD69 and CD25 expression on sh*FF* (control), sh*PTPN1*, sh*PTPN2* and sh*PTPN1*/sh*PTPN2* (dKD) cells with no stimulation or tumor-target K562 stimulation (E:T = 1:2) for 5 h. A representative figure of $n$ = three biological replicates, each tested in three technical replicates is shown. (F) A structural representation of the small molecule inhibitor L598, targeting both PTPN1 and PTPN2. (G–J) Flow cytometry analysis of CD25 and CD69 expression on L598 (10 µM) or vehicle control-treated NK-92 cells with or without tumor target K562 cell stimulations (E:T = 1:2) for 5 h. Representative flow cytometry blot from n = three biological replicates, each tested in three technical replicates are shown in (G) and (I). Pooled quantifications of geometric MFI for CD25 and CD69 expressions from $n$ = three biological replicates were presented in (H) and (J), respectively. *P*-values: (H) CD25: $p = 0.0002$ (***) at No Stimulation, $p = 0.0001$ (***) at E:T = 1:2 condition; (J) CD69: $p = 0.6573$ (ns) at No Stimulation, $p \leq 0.0001$ (***) at E:T = 1:2 condition (Two-way ANOVA with Šídák's multiple comparisons test). (K) Geometric MFI of CD122 (IL-2Rβ) was quantified by flow cytometry in NK-92 cells treated with L598 or vehicle control, with or without stimulation by K-562 tumor target cells. Pooled quantifications of $n$ = three biological replicates, each tested in three technical replicates. *P*-values: $p = 0.2266$ (ns) for no stimulation group; $p = 0.9941$ (ns) for E:T (1:2) (Two-way ANOVA with Šídák's multiple comparisons test). Source data are available online for this figure.

PTPN1 and PTPN2 are recognized as druggable targets in metabolic diseases and cancer with their inhibition in immune and tumor cells showing improved tumor growth control (Abdel-Magid, 2022; Baumgartner et al, 2023; Liang et al, 2023; Penafuerte et al, 2017). To evaluate whether pharmacological inhibition of PTPN1/N2 mimics the phenotype of dKD NK-92 cells, we used a small molecular inhibitor, L598 (Fig. 1F, Merck Frosst Inc.) (Montalibet et al, 2006), specific to both PTPN1/N2 catalytic domains without affecting their mRNA expressions (Fig. EV2B,C) (Penafuerte et al, 2017). Increasing the dosage of L598 (3.3 µM to 90 µM) demonstrated a clear dose-response enhancement of tumor cytolysis against K-562 cells (Fig. 2C) without affecting NK-92 cells' viability (Fig. EV2D). Consistently, a dose-dependent increase in tumor cytolysis was observed with KQ791 treatment, another dual-inhibitor derived from L598 (Figs. 2D and EV2E). Moreover, extending the dual-inhibitor treatment to 3 days further increased the fold change in cytolysis (Fig. 2B). Enhanced anti-tumor cytolysis was confirmed across various effector-to-target cell ratios, particularly against NK-sensitive K-562 tumors. In contrast, NK-resistant Reh cells, an acute lymphocytic leukemia line expressing MHC class I molecules, remained unaffected, underscoring the specificity of the cytotoxic activity (Fig. 2E; Appendix Fig. S1A,B). Acute myeloid leukemia (AML) is a clinically and genetically heterogeneous disease, representing one of the most challenging leukemia subtypes with the shortest 5-year survival rate (Kim and Choi, 2022; Rahmani et al, 2022; Xu and Niu, 2020). Treatment of NK-92 cells with the PTPN1/N2 dual inhibitor KQ791 significantly enhanced tumor cytolysis against all four tested AML cell lines: THP1, KG1A, MOLM14, and U937 AML cell lines (Fig. EV2F).

L598-treated NK-92 cells, similar to dKD cells, showed a significant upregulation of CD25 (Figs. 1G,H and EV2G) and CD69 (Figs. 1I,J and EV2H) reflected in both the proportion of positive cells and MFI, regardless of K-562 stimulation. In contrast, the IL-2 receptor β subunit (CD122) MFI was not significantly affected by pharmacological dual inhibition of PTPN1/N2 (Fig. 1K). Dual inhibition with L598 also significantly increased IFN-γ secretion by NK-92 cells, with or without tumor target cell stimulations (Fig. 2F). Notably, increased intracellular IFN-γ production was observed simultaneously with the upregulation of the degranulation marker LAMP-1 (also known as CD107a) on NK-92 cells (denoted by IFN-γ$^+$, CD107a$^+$ cells) during tumor target or PMA/Ionomycin

stimulations (Fig. 2G). The increased IFN-γ production during degranulation was associated with increased tumor target cell death, indicating that multiple anti-tumor responses could be enhanced through dual inhibition of PTPN1/N2 (Fig. 2H).

## PTPN1 and PTPN2 regulate the expression of cytolytic granules and alter their degranulation kinetics

After activation, NK cells quickly undergo cytoskeleton rearrangement, receptor polarization, cytolytic synapse formation, and direct exocytosis of cytolytic granules onto target cells (Mace et al, 2014). To understand the mechanism behind the enhanced anti-tumor cytolysis through dual inhibition of PTPN1/N2, we first evaluated the pre-synthesized cytolytic granules in NK-92 cells at the steady state. In humans, the granzyme family includes five members (A, B, H, K, and M). Interestingly, with the dual inhibitor treatment, the mRNA expression of granzyme B (GZMB), granzyme H (GZMH) and perforin (PRF1) but no other granzyme members were significantly upregulated compared with the vehicle control group (Fig. EV3A,B). Similarly, mRNA expression of the IFN-γ gene (IFNG) was upregulated (Fig. EV3A), corresponding to its increased protein production, as shown in Fig. 2F. These selective enhancement of pre-synthesized granzyme B and perforin expression but not granzyme A were also observed at the protein levels by flow cytometry (Fig. 2J) and western blot analyses (Fig. 2I) upon the dual inhibitor L598 treatment. Extending L598 treatment for 3 days also further increased the fold change in perforin and granzyme B expressions (Fig. EV3C).

Degranulation-mediated target cell death often occurs within minutes, as cytolytic granules are polarized via the microtubule-organizing centre (MTOC) and released at the immunological synapse through fusion with the plasma membrane (Mace et al, 2014). CD107a then becomes exposed on the NK cell membrane following degranulation, serving as a marker of this process and potentially contributing to self-protection against cytolytic granules (Mace et al, 2014; Prager and Watzl, 2019). After degranulation, NK cells enter a transient "resting" phase, during which they sense target cell death, restore cytolytic granules through new biogenesis and/or granules recycling, and subsequently detach to mediate serial killing (Li et al, 2011; Mace et al, 2014). Particularly, serial killing has been shown to follow burst kinetics, characterized by a

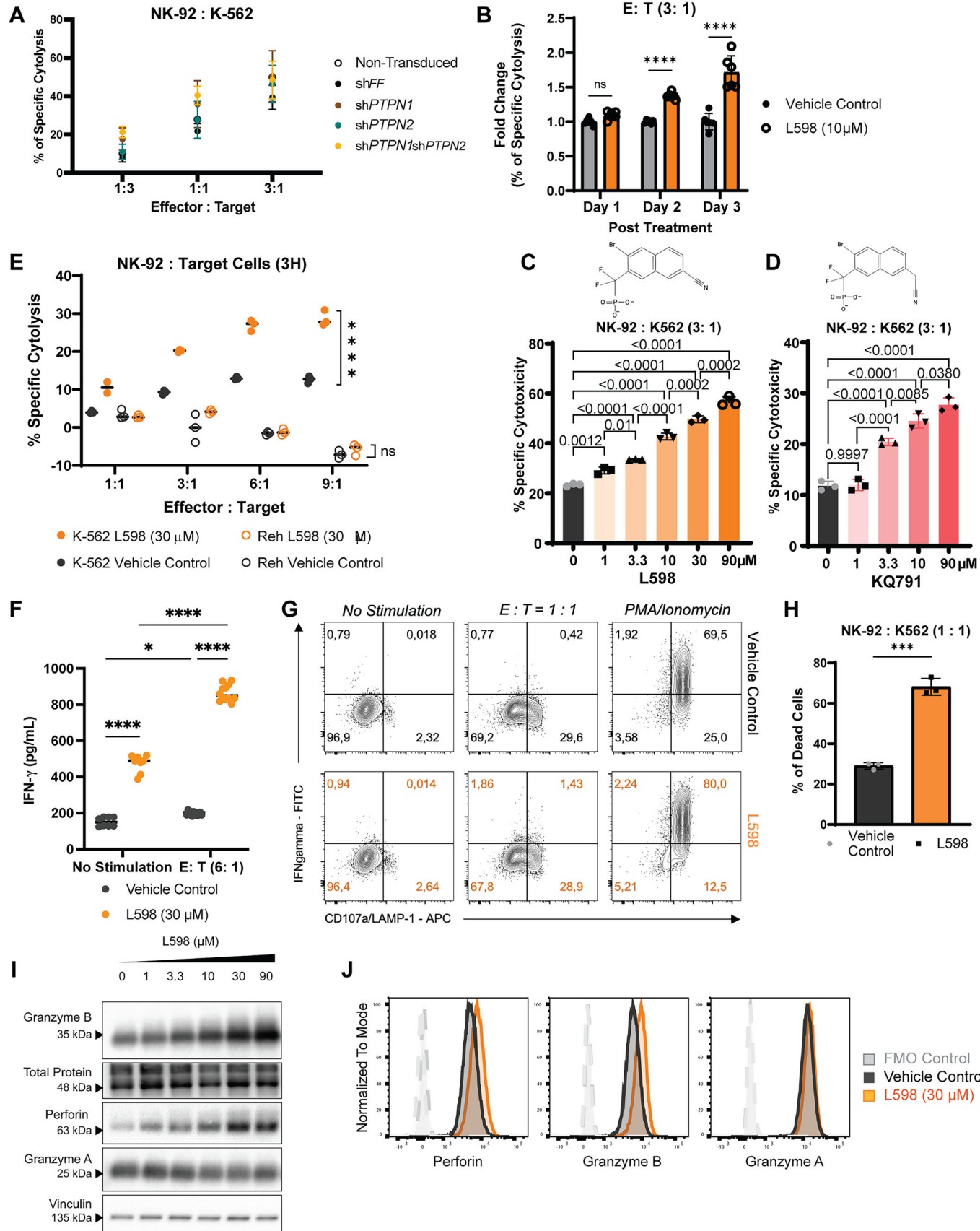

**Figure 2. PTPN1 and PTPN2 negatively regulate anti-tumor cytolytic functions of NK-92 cells.**

(A) Flow cytometry-based anti-tumor cytolysis assay of non-transduced, shFF, shPTPN1, shPTPN2, shPTPN1/shPTPN2 cells against tumor target K-562 cells at various effectors: target (E:T) cell ratios for 3 h. Results are from $n =$ two biological replicates at 1:3 (E:T) ratio and $n =$ three biological replicates at 1:1 and 3:1 (E:T) ratios. Data are presented as mean ± SD. (B) Fold change in specific cytolysis of K562 cells by NK-92 cells treated with L598 (10 µM) at an E:T ratio of 3:1, relative to vehicle control. NK-92 cells were pre-treated with L598 (10 µM) or vehicle control for 1, 2, or 3 days. After the indicated pre-treatment period, L598 was washed off, and NK-92 cells were then co-cultured with K562 cells for 4 h to assess cytolytic activity. Results are from two biological replicates; each tested in three technical replicates. Data are presented as mean ± SD. P-values: $p = 0.9335$ (ns) at Day 1; $p \leq 0.0001$ (****) at Day 2 and Day 3 (Two-way ANOVA with Šídák's multiple comparisons test). (C, D) Flow cytometry analysis of dose-dependent effects on K562 cell cytolysis by PTPN1/PTPN2 dual inhibitor L598 or KQ791-treated NK-92 cells after 3-h co-culture. Representative figures of $n =$ two biological replicates, each tested in three technical replicates (KQ791) and three or two technical replicates (L598) are shown. Data are presented as mean ± SD. P-values for L598: $p = 0.0012$ (**); $p = 0.01$ (*); $p = 0.0002$ (***); $p \leq 0.0001$ (****). P-values for KQ791: $p = 0.9997$ (ns); $p = 0.038$ (*); $p = 0.0085$ (**); $p \leq 0.0001$ (****) (One-way Anova with Tukey's multiple comparisons test). (E) Flow cytometry analysis on L598 (30 µM) treatment enhanced NK-92 cell's cytolysis against NK-sensitive K562 tumor cells and NK-resistant Reh tumor cells across various E:T ratios. A representative figure of $n = 7$ biological replicates (K562, each tested in three technical replicates) and $n = 3$ biological replicates (Reh, each tested in three technical replicates) is shown. P-values: $p = 0.0934$ (ns); $p \leq 0.0001$ (****) (Two-way ANOVA with Tukey's multiple comparisons test). (F) IFN-γ levels in L598 (30 µM) or vehicle treated NK-92 cell supernatants after 4-h co-culture with K562 cells (E:T = 6:1) or K562 media (no stimulation). Results are from $n =$ three biological replicates, tested in technical replicates of six per condition for stimulations and either six or two replicates for no stimulation controls. Values were normalized to IFN-γ secretion by K562 cells alone. P-values: $p = 0.0266$ (*); $p \leq 0.0001$ (****) (Two-way ANOVA with Tukey's test). (G, H) Flow cytometry analysis on IFN-γ production and CD107a expressions in NK-92 cells treated with (L598 (30 µM) or vehicle control following 6-h stimulation with K562 cells (E:T = 1:1), PMA/Ionomycin, or K562 media (no stimulation). A representative figure of $n =$ two biological replicates, each tested in three technical replicates is shown in (G). Percentages of dead K-562 cells under E:T = 1:1 from (G) are shown in (H). Data are presented as mean ± SD. P-value: $p = 0.0001$ (***) (unpaired t test) in (H). (I, J) protein expression of the intracellular cytolytic granules perforin, granzyme B, and granzyme A in steady-state NK-92 cells treated with L598 (30 µM) or vehicle, measured by western blot (I) and flow cytometry (J). Representative data from $n =$ two biological replicates (western blot) or $n =$ three biological replicates analyzed in duplicate or triplicate (flow cytometry) are shown. Source data are available online for this figure.

delayed first kill followed by more rapid subsequent kills (Choi and Mitchison, 2013).

To elucidate how elevated cytolytic effector molecules enhance NK cell cytolysis, we examined the kinetics of NK cell degranulation and anti-tumor killing. Tumor cell death was detectable as early as 15 min after co-culture with NK-92 cells, but a significant enhancement with dual PTPN1/N2 inhibition emerged at 60 min (Fig. 3A). Similarly, the cytolysis rate, which is calculated as the increase in dead tumor target cells over 1 h, was higher in L598-treated NK cells (slope = 0.2501) than in vehicle controls (slope = 0.1411) from 15 min onward, suggesting accelerated serial killings and an overall enhanced cytolysis (Fig. 3B). We further investigated NK cell degranulation dynamics and found that increased cytolysis was attributable to sustained degranulation activity over time (Fig. 3D). While perforin⁺ granzyme B⁺ NK cells declined significantly in vehicle-treated cells by 60 min, this population was maintained at higher levels in the L598-treated group at the same time point (Fig. 3D, E). The sustained presence of the perforin⁺, granzyme B⁺ population in the L598-treated group likely reflects more efficient replenishment of granzyme B required for serial killing, thereby supporting a more persistent cytolytic capacity (Fig. 3C). Remarkably, the rate of CD107a surface expression increased over time at similar levels in both the L598-treated and vehicle control groups, suggesting the kinetics of cytolytic granule release were not altered (Fig. 3F). However, whereas the degranulation rates of granzyme B and perforin gradually declined in vehicle-treated cells over the 1-h period, L598-treated NK-92 cells exhibited a transient increase in granzyme B degranulation rate at around 30 min, followed by a subsequent decline, consistent with burst killing (Fig. 3G). Notably, the perforin degranulation rate was initially delayed during the first 30 min but increased by 60 min in the L598-treated group (Fig. 3H).

Together, these observations suggest that dual inhibition of PTPN1 and PTPN2 enhances NK cell cytolysis through multiple mechanisms, including increased levels of pre-formed cytolytic effectors in the resting state, more efficient release of granzyme B

and perforin, and potential de novo synthesis of cytolytic granules to support persistent serial killing.

## Dual inhibition of PTPN1 and PTPN2 sensitizes NK cells to IL-2 receptor signaling

Since both PTPN1 and PTPN2 are major negative regulators of the JAK/STAT-mediated cytokine receptor signaling(Pike and Tremblay, 2016), we hypothesized that the inactivation of PTPN1/N2 enzymatic activities increases phosphorylation of transcription factor STATs, leading to transcriptional and translational expression of cytolytic effector molecules. Indeed, the dual inhibitor L598 showed a dosage-dependent increase in phosphorylation of STAT1 (Y701), STAT3 (Y705), STAT4 (Y694), and STAT5 (Y695) under standard cell culture conditions (Fig. 4A). In contrast, phosphorylation levels of STAT2 (Y690) and STAT6 (Y641) remained relatively stable (Fig. EV4A).

To determine whether increased tyrosine-phosphorylation of specific STAT members was driven by particular cytokine-receptor interaction(s) or was intrinsic to the inhibition of PTPN1/N2 phosphatases activities, we examined NK cell responses to two homeostatic cytokines IL-2 and IL-15, which are both essential for NK cell survival (Abel et al, 2018). Cytokines IL-2 and IL-15 activate both JAK1 (binding to CD122) and JAK3 (binding to CD132) but activate distinct STAT transcription factors. Short-term IL-2 but not IL-15 stimulations in L598 treated NK-92 cells led to enhance phosphorylation of these STAT members (Fig. 4B, lanes 3–12, Fig. EV4B), suggesting that the amplified STAT signaling by dual inhibition of PTPN1/N2 depends on extracellular IL-2 stimulation.

L598 treatment also increased serine phosphorylation of AKT (S473) but did not affect threonine and tyrosine phosphorylation of p38 MAPK (T180, Y182) in response to IL-2 or IL-15 (Appendix Fig. S2A–C). Moreover, genetic silencing of PTPN1 or PTPN2 similarly sensitized IL-2 receptor signaling, without further enhancing maximal signal strength but increasing basal phosphorylation levels of STAT3 (Fig. EV4C). Notably, protein expression of

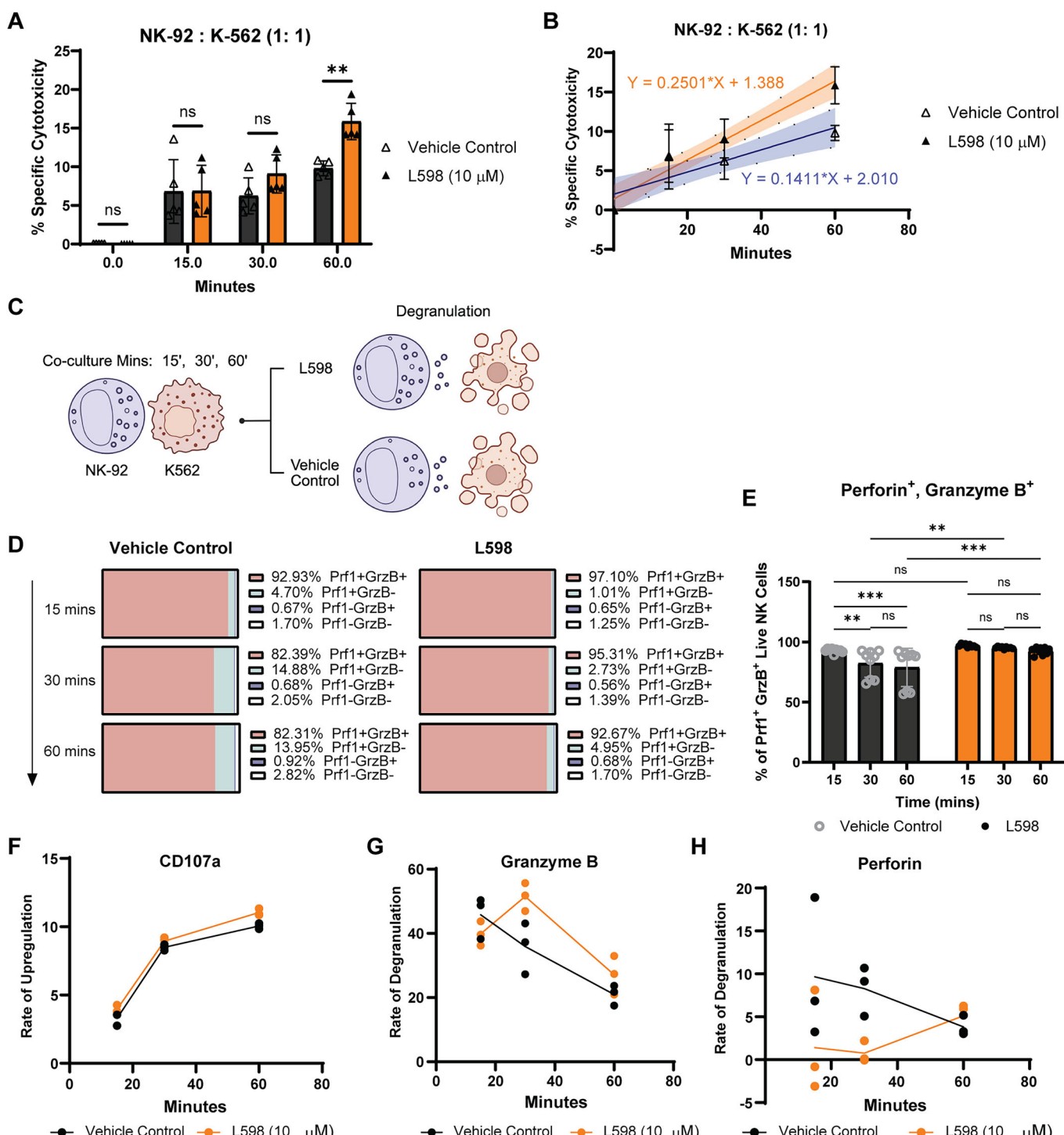

SOCS1, SOCS3, both are IL-2-induced negative feedback inhibitors, were only modestly increased by deficiency of PTPN1, PTPN2 or both, suggesting that PTPN1 and PTPN2 may primarily act during the early phase of the IL-2 response (Fig. EV4C).

We then verified that enhanced IL-2 receptor signaling resulting from dual PTPN1/N2 inhibition correlated with increased expression of the cytolytic proteins granzyme B and perforin, extending the upper limit of their transcriptional and translational expressions (Fig. 4C–F). No significant changes were observed in the expressions of T-BET and EOMES transcription factors with L598 treatment (Fig. EV4D,E). Together, these results suggested that dual inhibition of PTPN1/N2 activates STAT transcription factors, driving to higher expressions of granzyme B and perforin during IL-2 receptor signaling.

◄ **Figure 3. Dual inhibition of PTPN1 and PTPN2 alters cytolytic activity and degranulation kinetics of NK-92 cells.**

(A, B) Time-course analysis of L598 (10 μM) or vehicle control-treated NK-92-mediated cytolysis of K562 tumor cells at an E:T ratio of 1:1, measured at 15, 30, and 60 min. Results are from $n$ = three biological replicates, each tested in two or three technical replicates. Data are presented as mean ± SD. $P$-values: $p$ = 0.0014 (**); $p$ > 0.1234 (ns) (Two-way ANOVA with Šídák's multiple comparisons test) in (A). In (B), simple linear regression test was used to compare the cytolysis rates between the L598 treatment and control groups. Slope: vehicle control = 0.1411; L598 = 0.2501 ($P$-value: $p$ = 0.0082). (C) Schematic representation of L598, a dual PTPN1/PTPN2 inhibitor, enhancing NK cell degranulation efficiency over 60 min. (D, E) Flow cytometry analysis of intracellular perforin (Prf1) and granzyme B (GrzB) in NK-92 cells after 15, 30, and 60 min of co-culture with K562 cells (E:T = 1:1). Mean percentages of Prf1⁺GrzB⁺, Prf1⁺GrzB⁻, Prf1⁻GrzB⁺, and Prf1⁻GrzB⁻ populations were plotted, with Prf1⁺GrzB⁺ cells quantified. Results are from $n$ = three biological replicates, each tested in three technical replicates. Data are presented as mean ± SD. $P$-values: $p$ = 0.0185 (*) for vehicle (15 min) vs. vehicle (30 min); $p$ = 0.0057 (**) for vehicle (30 min) vs. L598 (30 min); $p$ = 0.0014 (**) for vehicle (60 min) vs. L598 (60 min); $p$ > 0.1234 (ns) (Two-way ANOVA with Šídák's test). (F–H) Flow cytometry analysis of CD107a surface expression (MFI upregulation) rate and degranulation rates (MFIs) of granzyme B and perforin during the 60-minute co-culture. Representative data from $n$ = two biological replicates (CD107a) and $n$ = three biological replicates (granzyme B and perforin), each tested in two or three technical replicates are shown. Source data are available online for this figure.

## Sensitized IL-2 responses in NK cells are partially resistant against immunosuppressive TGFβ1

In the tumor microenvironment, cytokines such as TGFβ1, IL-35 and IL-4 suppress NK cell activities (Mirlekar, 2022; Picant et al, 2025). We investigated whether dual inhibition of PTPN1/N2 sensitizes NK-92 cells to these immunosuppressive cytokines. Treatment with TGFβ1 (1 ng/mL) alone or combined with IL-35 (10 ng/mL of IL-35) or IL-4 (10 ng/mL) significantly reduced NK-92 cell anti-tumor cytolysis whereas IL-4 (10 ng/mL) or IL-35 (10 ng/mL or 20 ng/mL) alone had a minimal effect (Fig. 5A). Under various immunosuppressive conditions (TGFβ1 alone, or in combination, with or without IL-4 or IL-35), the L598 treatment group exhibited significantly stronger tumor cell elimination than the control group (Fig. 5B). The enhanced anti-tumor cytolytic response was consistently observed across various effector-to-target cell ratios with TGFβ1 or IL-4 treatment in the L598 group over the control group (Fig. 5C; Appendix Fig. S3A). Despite TGFβ1 reducing granzyme B and perforin expression, L598-treated NK cells retained significantly higher amounts of granzyme B (Fig. 5D) and slightly more perforin (Appendix Fig. S3B) than the control group.

Next, we performed cell signaling analysis to investigate the molecular mechanisms underlying the partial resistance against TGFβ1-mediated suppression. Although TGFβ1 treatment upregulated serine phosphorylation of SMAD2 (S465/467) in both the dual inhibitor L598 treatment and vehicle control groups (Appendix Fig. S3C), higher levels of phosphorylated STAT1 (Y701), STAT3 (Y705), and STAT4 (Y693) and slightly STAT5 (Y694) were maintained in the L598 cells (Fig. 5F).

To further enhance resistance to TGFβ1-mediated suppression, we combined soluble TGFβ1 receptor II (sTβRII) with dual inhibition of PTPN1/N2. sTβRII alone did not affect NK-92 cell cytotoxicity, but its addition modestly improved cytolysis both with and without L598 treatment (Fig. 5E). The combination further restored cytolysis in TGFβ1-suppressed, L598-treated cells; however, it did not fully recover to the levels observed in non-suppressed, L598-treated cells (Fig. 5E).

## Genetic dual silencing and pharmacological dual inhibition of PTPN1 and PTPN2 enhance anti-tumor responses in vivo

We verified that dual inhibition of PTPN1/N2 enhanced NK-92 cell cytolysis against eGFP-expressing solid tumor cell lines: Caki-1

(human renal cell carcinoma, Fig. EV5A,B), U87 (human glioblastoma,), and P3 (glioblastoma patient derived cells) (Fig. EV5C–F) through real-time IncuCyte analysis where tumor cell death was quantified by decreased eGFP signal. In P3 cytolysis assay, a red emission caspase 3/7 cleavage reporter dye was added to facilitate distinction of NK-92 cells from P3 glioblastoma spheroids. In U87 cytolysis assay, treatment with L598 alone did not significantly delay tumor growth or induce tumor cell death, indicating anti-tumor effect requires NK cell activity (Fig. EV5I).

To assess NK cell anti-tumor responses in vivo, PTPN1/N2-targeted NK-92 cells or control cells were adoptively transferred into male immunodeficient NOD-*scid* IL2Rγ^null^ (NSG) mice bearing subcutaneous U87 tumors (isolated from a male patient). The control sh*FF* cell treatment group ($n$ = 3) significantly reduced tumor size after three doses compared to the cell-free vehicle treatment (denoted as No Treatment) group ($n$ = 7) ($p$-value = 0.0081), while the dKD cell treatment ($n$ = 3) further delayed tumor progression ($p$-value ≤0.0001 compared with sh*FF* control group; $p$-value = 0.0384 compared with No Treatment group) (Fig. 6A,B). Pre-treating control sh*FF* NK-92 cells with the PTPN1/N2 dual inhibitor L598 prior to adoptive transfer resulted in the smallest tumor size ($n$ = 5; $p$-value ≤0.0001 compared with No Treatment group; $p$-value = 0.0182 compared with sh*FF* control group) after nine doses compared to the vehicle control-sh*FF* group ($n$ = 7) or No Treatment group ($n$ = 8) (Fig. 6C,D). Control or L598-treated sh*FF* NK-92 treatment did not significantly alter body weight of healthy or U87-bearing NSG mice (Appendix Fig. S6).

Metastatic triple-negative breast cancer (TNBC) is the most aggressive subtype of breast cancer, with limited treatment options currently available, including surgery, adjuvant chemotherapy, and radiotherapy (Li et al, 2022). Given these considerations, we investigated whether dual inhibition of PTPN1/N2 could enhance NK cell therapy for patients with TNBC. Using the MDA-MB-231 human cell line as a TNBC model, we observed that these cells were susceptible to NK-92 cell-mediated cytolysis in vitro (Fig. EV5G,H). Pre-treating NK-92 cells with the dual-inhibitor of PTPN1/N2 further increased their cytolytic activity against MDA-MB-231 cells (Fig. EV5G,H). To assess the therapeutic potential in vivo, we orthotopically injected MDA-MB-231 TR (triple reporter) cells into the mammary fat pads of female NSG mice and administered dual-inhibitor KQ791-pre-treated or control NK-92 cell therapy twice per week (Fig. 6E). After three doses, mice receiving dual-inhibitor-enhanced NK-92 cell therapy ($n$ = 5) exhibited a reduction in tumor volume compared to those receiving control NK-92 cell therapy ($n$ = 4; $p$-value ≤0.0001), as measured by calipers (Fig. 6F). In

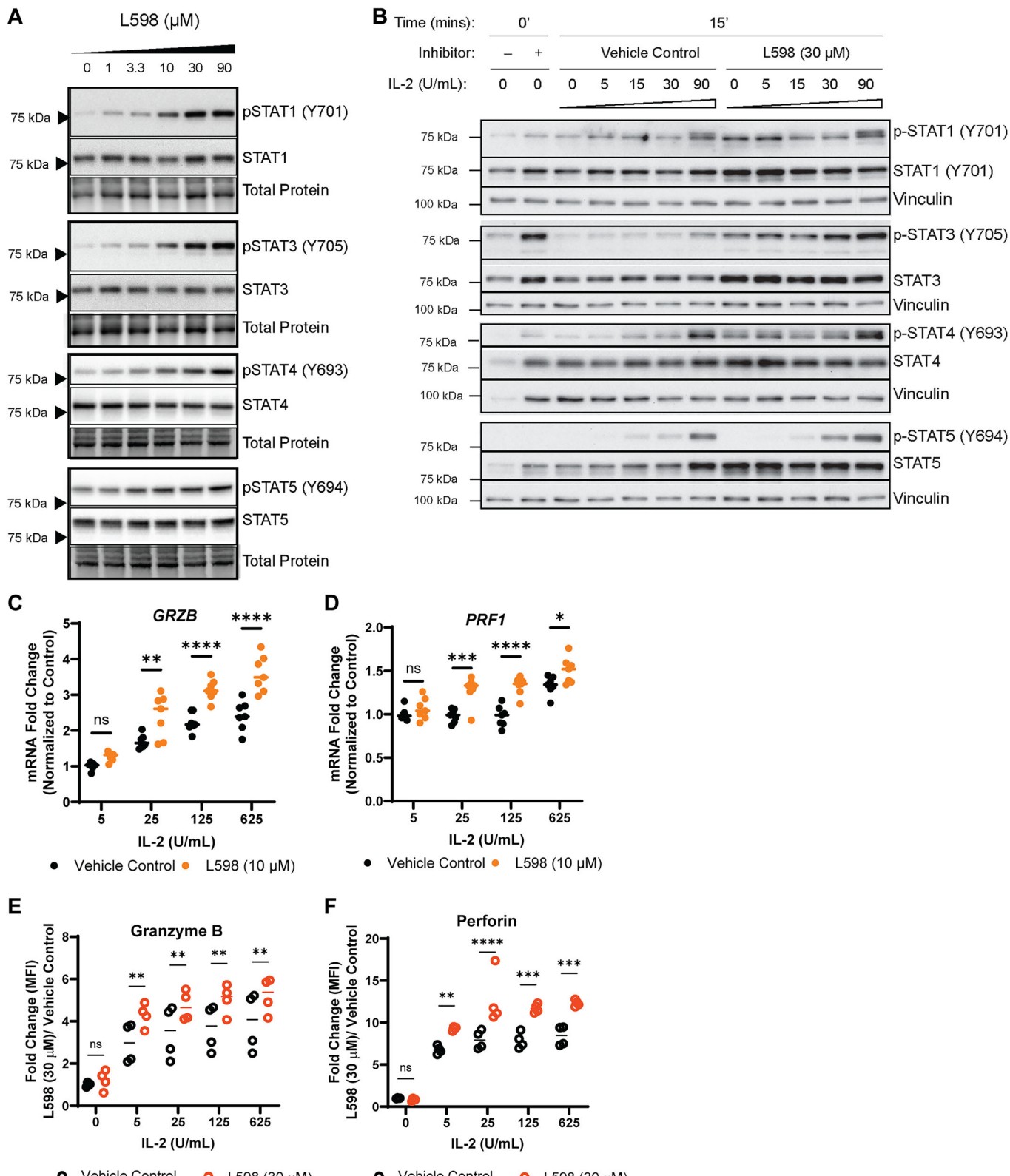

◄ **Figure 4.  Dual inhibition of PTPN1 and PTPN2 sensitizes NK-92 cells to IL-2 receptor signaling, enhancing granzyme B and perforin expression.**

(**A**) Dosage-dependent STAT signaling in NK-92 cells treated with an increasing concentration of L598 in standard cell culture conditions supplemented with IL-2 (100 IU/mL) for 3 days. A representative figure from $n$ = two biological replicates is shown. Total protein ( ~ 48 kDa) serves as the loading control. (**B**) Western blot analysis of IL-2 activated STAT signaling in NK-92 cells pre-treated with L598 (30 μM) or vehicle control, following 6-h serum starvation and 15-minute stimulation with increasing IL-2 concentrations. A representative figure from n = two biological replicates is shown. The same membrane was sequentially probed for (p)-STAT3 and (p)-STAT5; therefore, the same vinculin loading control is shown for both. (**C, D**) qRT-PCR analysis of cytolytic granules granzyme B (*GZMB*) and perforin (*PRF1*) in NK-92 cells cultured for 3 days with L598 (10 μM) or vehicle control under increasing IL-2 concentrations (5–625 IU/mL). Results are from $n$ = two biological replicates, each tested in three or four technical replicates. *P*-values (*GZMB*): $p$ = 0.5312 (ns), $p$ = 0.0015 (**), $p \leq 0.0001$ (****) (Two-way ANOVA with Šídák's test); *P*-values (*PRF1*): $p$ = 0.8058 (ns), $p$ = 0.0388 (*), $p$ = 0.0001 (***), $p \leq 0.0001$ (****) (Two-way ANOVA with Šídák's test). (**E, F**) flow cytometry analysis of intracellular granzyme B and perforin in NK-92 cells treated with L598 (30 μM) or vehicle control, following 3-day culture with increasing IL-2 concentrations (0–625 IU/mL). Results are from $n$ = two biological replicates, each tested in technical replicates of two. *P*-values (granzyme B): $p$ = 0.9641 (ns), $p$ = 0.0028 (**) at 5 IU/mL, $p$ = 0.0041 (**) at 25 IU/mL, $p$ = 0.0022 (**) at 125 IU/mL, $p$ = 0.0045 (**) at 625 IU/mL (Two-way ANOVA with Šídák's test); *P*-values (perforin): $p$ = 0.9996 (ns), $p$= 0.086 (**), $p$ = 0.0002 (***), $p \leq 0.0001$ (****) (Two-way ANOVA with Šídák's test). Source data are available online for this figure.

contrast, tumor sizes in the non-treatment ($n$ = 6) and control NK-92 cell therapy groups remained comparable throughout the study, suggesting that NK cell therapy alone is ineffective against the MDA-MB-231 TR tumor model in vivo but can be improved by dual inhibition of PTPN1/N2 (Fig. 6F).

## Translational implications of dual targeting PTPN1 and PTPN2 to improve umbilical cord blood NK therapy

Umbilical cord blood has been a favored source for adoptive cell transplantation because of the lower risk of developing GvHD and the feasibility of developing "off-the-shelf" therapies (Zhao et al, 2020). To assess the translational potential of our findings in enhancing CB NK cell therapies, we treated ex vivo-expanded CB NK cells with the dual inhibitor L598 for 5 days. The CB NK cells were derived from optimal cord blood units, defined as those cryopreserved within 24 h of collection and with a nucleated red blood cell (NRBC) count $\leq 8 \times 10^7$ (Marin et al, 2024). Like NK-92 cells, CB NK cells that received L598 treatment expressed significantly greater amounts of perforin and granzyme B but not granzyme A (Fig. 7A–C). The percentage of CD69$^+$ cell populations and the MFI of CD69 were also upregulated with tumor target K-562 stimulations in two independent NK cell donors (Fig. 7D,E).

We then examined the anti-tumor cytolytic responses of CB NK cells. Overall, the pooled analysis across all 15 donors demonstrated significant improvement following L598 treatment, as assessed by paired t-test (Fig. 7F). However, there were notable variations in cytolytic potency across donors. Specifically, CB NK cells from 11 donors exhibited varying degrees of enhanced cytolysis against K-562 cells after L598 treatment, whereas CB NK cells from four donors were less sensitive or unresponsive to the inhibitor (Fig. 7G). This functional heterogeneity may arise from differences in NK cell subsets. Therefore, we examined CB NK subsets based on CD56 and CD16 expression. Notably, the proportions of CD56$^{bright}$, CD16$^+$ and CD56$^{bright}$, CD16$^-$ were varied considerably across the 15 donors. In addition, the expression levels of selected germ-line-encoded receptors, such as NKp30 on CD56$^{bright}$, CD16$^-$ cells, also showed a broad range of inter-donor variability (Fig. 7H,I). In CB NK cells from two particularly responsive donors, L598 treatment further enhanced their anti-tumor cytolysis against glioblastoma patient-derived eGFP$^+$ P3 cells compared with the vehicle control group (Fig. 7J,K).

Taken together, our results suggest that dual targeting of PTPN1/N2 by genetic silencing or dual pharmacological inhibition

may enhance human NK cell-based immunotherapy for the treatment of immunosuppressive cancers such as glioblastoma.

## Discussion

The cytokine milieu shapes NK cell anti-tumor responses, influencing cytolytic functions, differentiation, maturation, and antigen-specific "memory" formation (Gotthardt et al, 2019; Wu et al, 2017). IL-2 and IL-15 directly enhance NK cell cytolytic functions or prime responses to other cytokines, like IL-12, IL-18, IL-21, and type I IFN (Nielsen et al, 2016; Wang et al, 2000; Wu et al, 2017). Our results showed that dual inhibition of PTPN1/N2 in NK-92 cells upregulated phosphorylation of STAT1, STAT3, STAT4 and STAT5 in response to IL-2 stimulation (Fig. 4A,B). Among these activated STAT transcription factors, STAT1, STAT4, and STAT5, are known to enhance NK cell anti-tumor cytolytic functions, IFN-γ production (STAT1 and STAT4), CD25 upregulation (STAT4 and c-Jun) and NK cell survival, maturation, and cytolytic functions (STAT5) (Gotthardt and Sexl, 2016; Gotthardt et al, 2019; O'Sullivan et al, 2004). In contrast, STAT3 plays a controversial role. In NK-92 cells, STAT3 is positively linked to proliferation but negatively associated with the expression of activating receptors and cytolytic effector molecules (Gotthardt and Sexl, 2016; Witalisz-Siepracka et al, 2022). A study demonstrated that inhibiting STAT3 and blocking the STAT3-activating cytokine IL-6 reduced the expression of inhibitory receptor TIGIT in NK-92 cells (Gonzalez-Ochoa et al, 2022). While the role of activated STAT3 remains unclear in our studies, targeting STAT3 combination with dual inhibition of PTPN1/N2 may help sustain NK cell cytolytic functions and prevent dysfunction.

Dual inhibition of PTPN1/N2 sensitized NK cells to low-dose IL-2 stimulation (10 or 30 IU/mL), potentially through CD25 upregulations, leading to increased expressions of the cytolytic granules granzyme B and perforin in both NK-92 cells and cord blood NK cells, consistent with a previous report (Zhang et al, 2008). CD25 upregulation on NK cells was evident in the resting state and persisted upon tumor target cell stimulations in both sh*PTPN1* and sh*PTPN2* single or double knockdown cells (Figs. 1D and EV1G,H) as well as with pharmacological dual inhibition of PTPN1/N2 (Fig. 1G,H). These findings suggest that NK cells with dual targeting of PTPN1/N2 may better compete with regulatory T (Treg) cells, which constitutively express CD25, for IL-2 binding. Sensitizing the IL-2 response through dual inhibition of

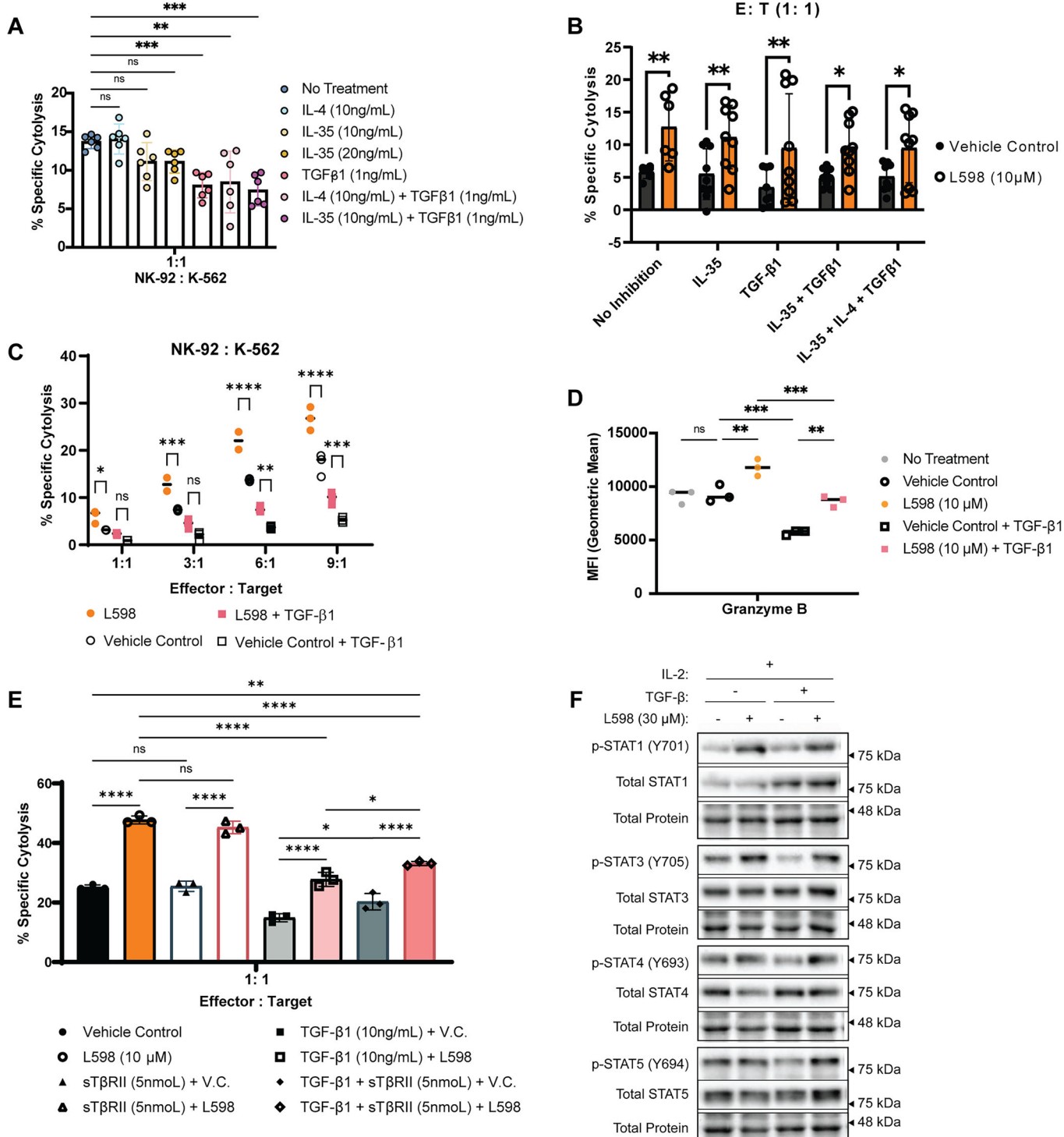

PTPN1/N2 could eliminate the need for exogenous IL-2 or IL-15 administration to maintain adoptively transferred NK cell survival in cancer patients, potentially preventing severe side effects (Christodoulou et al, 2021; Felices et al, 2018; Raeber et al, 2023).

Targeting PTPN1 and PTPN2 concurrently potentiated NK cell cytolytic responses and conferred resistance to immune suppression mediated by the inhibitory cytokine TGFβ-1, which acts

through the Smad2/3/4 signaling pathway (Fig. 5B,C). Resistance to TGFβ-1-mediated suppression may occur through crosstalk among signal transducers or synergistic effects by transcription factors. Signal transducer Smad3 is downstream of the TGFβ-1 receptor and its activities can be restricted by PI3K/Akt in various cell types (Guo and Wang, 2009). Our results showed that dual inhibitor-treated NK-92 cells increased phosphorylation of AKT (S473) in

◄ **Figure 5. Sensitized IL-2 response in NK cells are partially resistant to immunosuppressive TGFβ1.**

(A) Suppression of NK-92 cell cytolysis by cytokines: IL-35 (20 ng/mL), TGFβ1 (1 ng/mL) alone, or TGFβ1 (1 ng/mL) and IL-4 (10 ng/mL) or TGFβ1 (1 ng/mL) and IL-35 (10 ng/mL) in combination. Results are from two biological replicates; each tested in three technical replicates. $P$-values: $p = 0.0044$ and $p = 0.0037$ (**), $p = 0.0002$ (***), $p \leq 0.0001$ (****) (One-way ANOVA with Dunnett's test). (B) NK-92 cells treated with L598 (10 µM) showed enhanced cytolysis against K562 cells compared to vehicle controls, despite inhibitory cytokine suppression. Results are from $n =$ two biological replicates for the "no inhibition" group and $n =$ three biological replicates for each inhibition group; each tested in three technical replicates. $P = 0.0021$ (**); $p = 0.0140$ (*) for IL-35 + TGF-β1; $p = 0.0153$ (*) for IL-35 + IL-4 + TGF-β1 (Two-way ANOVA with Šídák's multiple comparison test). (C) NK-92 cells treated with TGFβ1 (10 ng/mL) and either L598 (10 µM) or vehicle control were assessed for cytolysis against K-562 cells across varying E:T ratios. Data were from $n =$ two biological replicates, each tested in three technical replicates, is shown as a representative scatter plot with the median indicated. $P$-values (E:T = 1:1): $p = 0.3995$ (ns) with TGF-β1 treatment, $p = 0.0396$ (*) without TGF-β1 treatment; $P$-values (E:T = 3:1): $p = 0.0746$ (ns) with TGF-β1 treatment, $p = 0.0003$ (***) without TGF-β1 treatment; $P$-values (E:T = 6:1): $p = 0.0036$ (**) with TGF-β1 treatment, $p \leq 0.0001$ (****) without TGF-β1 treatment; $P$-values (E:T = 9:1): $p = 0.0005$ (***) with TGF-β1 treatment, $p \leq 0.0001$ (****) without TGF-β1 treatment (Two-way ANOVA with Tukey's test). (D) Intracellular granzyme B levels in NK-92 cells treated with L598 (10 µM) or vehicle control in the presence of TGFβ1 (10 ng/mL) were analyzed by flow cytometry. Results are from $n =$ two biological replicates, each tested in three technical replicates, is shown as a representative scatter plot with the median indicated. $P$-values: $p = 0.9934$ (ns), $p = 0.0028$ (**), $p = 0.0014$ (**), $p = 0.0002$ (***), $p = 0.0004$ (***) (One-way ANOVA with Tukey' test). (E) Flow cytometry analysis of anti-tumor cytolysis by NK-92 cells treated with L598 (10 µM) or vehicle control under TGF-β1-mediated suppression (10 ng/mL), with or without sTβRII (5 nM) blockade. A representative figure of $n =$ two biological replicates, each tested in three technical replicates is shown. Data are presented as mean ± SD. $P$-values: $p = 0.0335$ (*) for TGF-β1 + sTβRII + L598, $p = 0.0309$ (*) for TGF-β1 + vehicle control v.s. TGF-β1 + sTβRII + vehicle control, $p = 0.001$ (**) for vehicle control v.s. TGF-β1 + sTβRII + L598, $p \leq 0.0001$ (****) (One-way ANOVA with Tukey's multiple comparisons test). (F) Western blot analysis of STAT signaling in NK-92 cells treated with L598 (30 µM) or vehicle control in the presence of TGFβ1 (10 ng/mL). A representative figure of $n =$ three biological replicates is shown. Source data are available online for this figure.

response to IL-2, which may antagonize Smad3-mediated inhibitory activities (Appendix Figs. S2A,C and S3C). Furthermore, TGFβ1 exhibits context-dependent regulation of JAK/STAT signaling (Luo, 2017). In T cells, TGFβ-1 inhibits IL-12-induced phosphorylation of STAT3 and STAT4 (Luo, 2017). In contrast, our results showed that dual inhibitor-treated NK92 maintained increased phosphorylation of STAT1, STAT3 and STAT4 upon TGFβ1 exposure (Fig. 5F), suggesting that the observed partial resistance may arise from events occurring upstream of these phosphorylated STAT proteins.

PTPN2 negatively regulates TGFβRII through binding to receptor integrin α1β1 (Chen et al, 2014). Interestingly, an alternative scenario is found in PTPN1-deficient liver cells, where there is resistance to TGFβ1. This response is accompanied by an increased expression of SnoN (inhibitor of TGFβ1 signaling) and an activation of NF-κB, while phosphorylation of SMAD2/3 was upregulated (Ortiz et al, 2012; Revuelta-Cervantes et al, 2011). These results suggest that PTPN1 and PTPN2 may play contradictory roles in regulating TGFβ1 receptor signaling. Considering the higher abundance of PTPN1 compared to PTPN2 in NK cells (Fig. 1A,B), we suspect that PTPN1 inhibition–mediated resistance to TGFβ1 signaling may override the enhancing effect on TGFβ1 signaling caused by PTPN2 inhibition.

PTPN1 and PTPN2 are druggable protein tyrosine phosphatases in treating human diseases including obesity, diabetes, and cancer (Liu et al, 2022; Loh et al, 2011). In this study, we utilized the synthetic small molecule inhibitors L598 (Merck Frosst Inc. in the public domain) (Montalibet et al, 2006) and its close derivative, KQ791, which is in phase I clinical trials (NCT02445911, NCT02370043) for treating type-2 diabetes mellitus. Both L598 and KQ791 are derived from the DFMP inhibitor 7-bromo-6-difluoromethylphosphonate 3-naphthalenenitrile (Kanyr Inc.) (Montalibet et al, 2006), and selectively target the enzymatic sites of PTPN1/PTPN2 as reported in enhancing dendritic cell-based therapy (Penafuerte et al, 2017). In particular, the phosphonate group of these dual inhibitors establishes multiple hydrogen bonds with the PTP catalytic loop (Figs. 1F and EV2E). Additionally, the two fluorine atoms interact with the phenyl side chain of Phe182 via van der Waals forces and establish hydrogen bonds, mediated

by a water molecule, with the side chain nitrogen of Gln266 and the main chain nitrogen of Phe182. The naphthalene ring is positioned between Tyr46 and Phe182, while the nitrile group extends toward the secondary binding site (Montalibet et al, 2006).

By treating NK-92 cells and CB NK cells ex vivo with these PTPN1/N2 dual-inhibitor, we observed enhanced anti-tumor cytolytic activities. Dual inhibitors of PTPN1/N2 are being explored for treating various solid tumors that are partially responsive to anti-PD-1 immune checkpoint inhibitor therapy in pre-clinical models and clinical trials (e.g., AC484 in NCT04777994), led by academic and industrial labs (AbbVie Inc., and Kanyr Pharma) (Baumgartner et al, 2023; Liang et al, 2023; Perez-Quintero et al, 2025). Systemic injections of these dual inhibitors demonstrate efficacy in repressing tumor growth, sensitizing responses to anti-PD1 therapy, and promoting tumor infiltration of effector lymphocytes, along with expansions of granzyme B high CD8 + T cells and NK cells (Baumgartner et al, 2023; Liang et al, 2023). However, systemic inhibition may not be suitable for cancers with hyperactivated STAT transcription factors or resistance to dual inhibition and IFN-γ-mediated growth inhibition (e.g., ganglioneuromas and endometrium cancer) (Baumgartner et al, 2023). Therefore, we employed ex vivo dual inhibition of PTPN1/N2 to enhance NK cells anti-tumor responses against hard-to-treat cancers, such as glioblastoma cell lines and triple-negative breast cancer cell lines (Fig. 6).

In this study, we utilized CB NK cells collected from optimal cord blood unit (Marin et al, 2024). However, we observed notable inter-donor variability in both baseline cytolysis and responsiveness to dual inhibition of PTPN1/N2 (Fig. 7G). This variability may be attributed to genetic polymorphisms, such as single-nucleotide polymorphisms (SNPs) at specific cytokine (e.g., *Il1b, Il2,* and *Il12b*) and cytokine receptor loci (e.g., *IL10RB, IL2RG*), which have been associated with individual differences in immune function (Hoang Nguyen et al, 2025; Vakil et al, 2022). Given the observed correlation between enhanced NK cell cytolysis and activated IL-2 signaling, we further examined the expression of IL-2 receptor subunit across donors (Fig. 7I). NK cell heterogeneity has been well-documented in peripheral blood, bone marrow and tissues, driven by diversity in germ-line encoded receptors (Crinier et al,

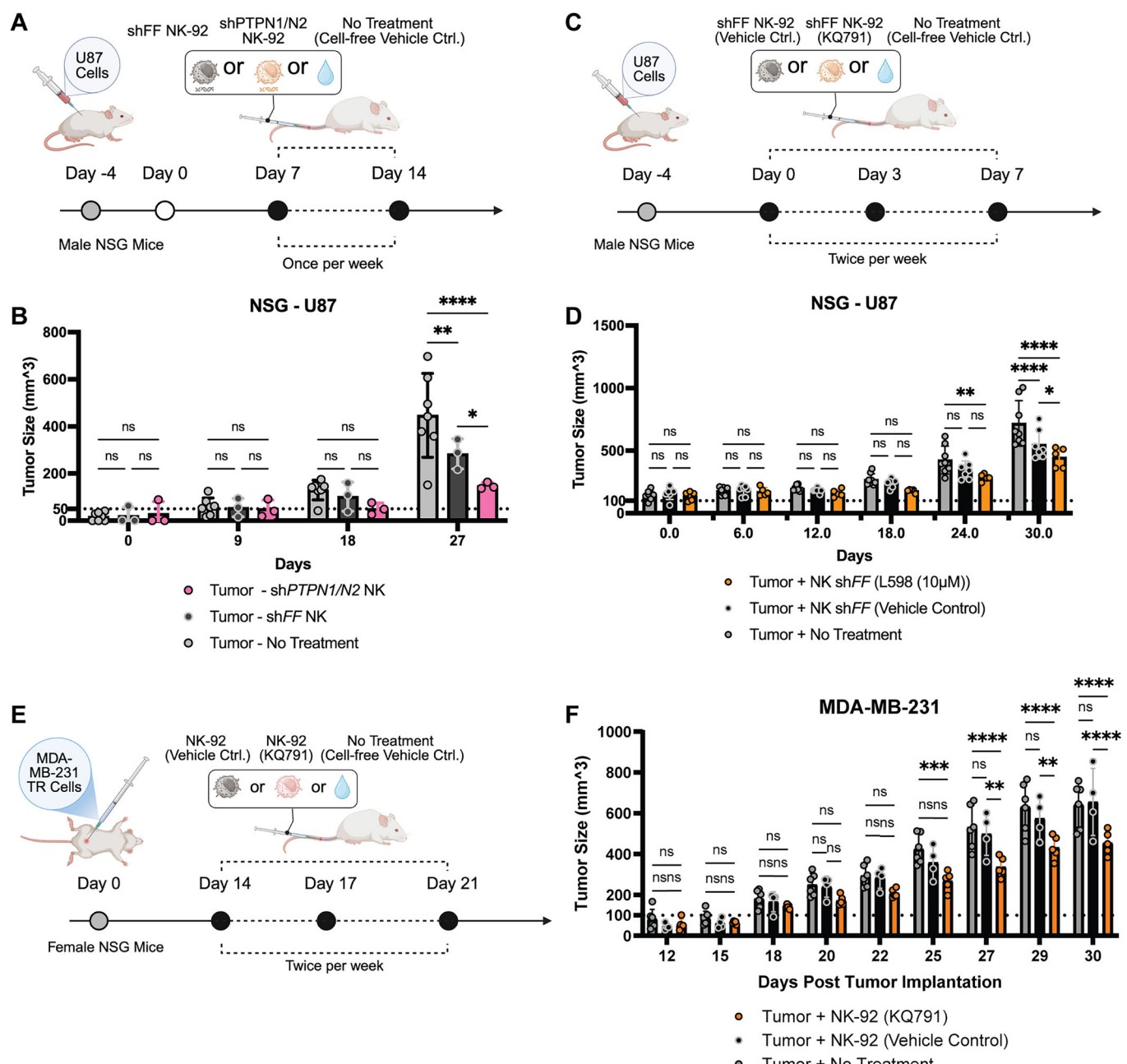

**Figure 6. Dual targeting of PTPN1 and PTPN2 enhances NK-92 cell anti-tumor growth control of glioblastoma or TNBC cells in xenograft NSG models.**

(A) Schematic representation of the in vivo experimental design involving genetic perturbation, in which male NSG mice bearing subcutaneous human U87 glioblastoma tumors received adoptive transfer of NK-92 cells expressing shFF or shPTPN1/PTPN2. (B) Tumor volume in NSG mice treated once weekly for 3 weeks with NK-92 cells expressing shFF ($n = 3$ biological replicates), shPTPN1/PTPN2 ($n = 3$ biological replicates), or receiving no treatment ($n = 7$ biological replicates). Data are presented as mean ± SD. P-values: $p > 0.05$ (ns), $p = 0.0384$ (*), $p = 0.0081$ (**), $p \leq 0.0001$ (****) (Two-way ANOVA with Holm-Šídák test). (C) Schematic representation of the in vivo experimental design involving pharmacological dual inhibition, in which male NSG mice bearing subcutaneous human U87 glioblastoma tumors received adoptive transfer of shFF-expressing NK-92 cells pre-treated with a dual inhibitor (L598, 10 μM) or vehicle control or received cell-free vehicle control ("no treatment"). (D) Tumor volume in NSG mice treated twice weekly for 4.5 weeks with L598-treated shFF NK-92 cells ($n = 5$ biological replicates), vehicle-treated shFF NK-92 cells ($n = 7$ biological replicates), or no treatment ($n = 8$ biological replicates). Data are presented as mean ± SD. P-values: $p > 0.05$ (ns), $p = 0.0182$ (*), $p = 0.0033$ (**), $p \leq 0.0001$ (****) (Two-way ANOVA with Holm-Šídák test). (E) A schematic representation of the in vivo experiment evaluating KQ-791-treated NK-92 cell therapy. Female NSG mice bearing orthotopic MDA-MB-231TR mammary tumors received twice-weekly adoptive transfers of NK-92 cells pre-treated with KQ-791 (10 μM), vehicle control, or no cells ("No Treatment") over a 2-week period. (F) Tumor volume in NSG mice treated with KQ791-NK-92 cells ($n = 5$ biological replicates), or vehicle-NK-92 cells ($n = 4$ biological replicates) or receiving no treatment ($n = 6$ biological replicates). Data are presented as mean ± SD. P-values: $p > 0.05$ (ns), $p = 0.002$ (**) on Day 27 KQ791 vs vehicle control, $p = 0.0075$ (**) on Day 29 KQ791 vs vehicle control, $p = 0.009$ (***) on Day 25 KQ791 vs no treatment, $p \leq 0.0001$ (****) (Two-way ANOVA with Tukey's multiple comparisons test). Source data are available online for this figure.

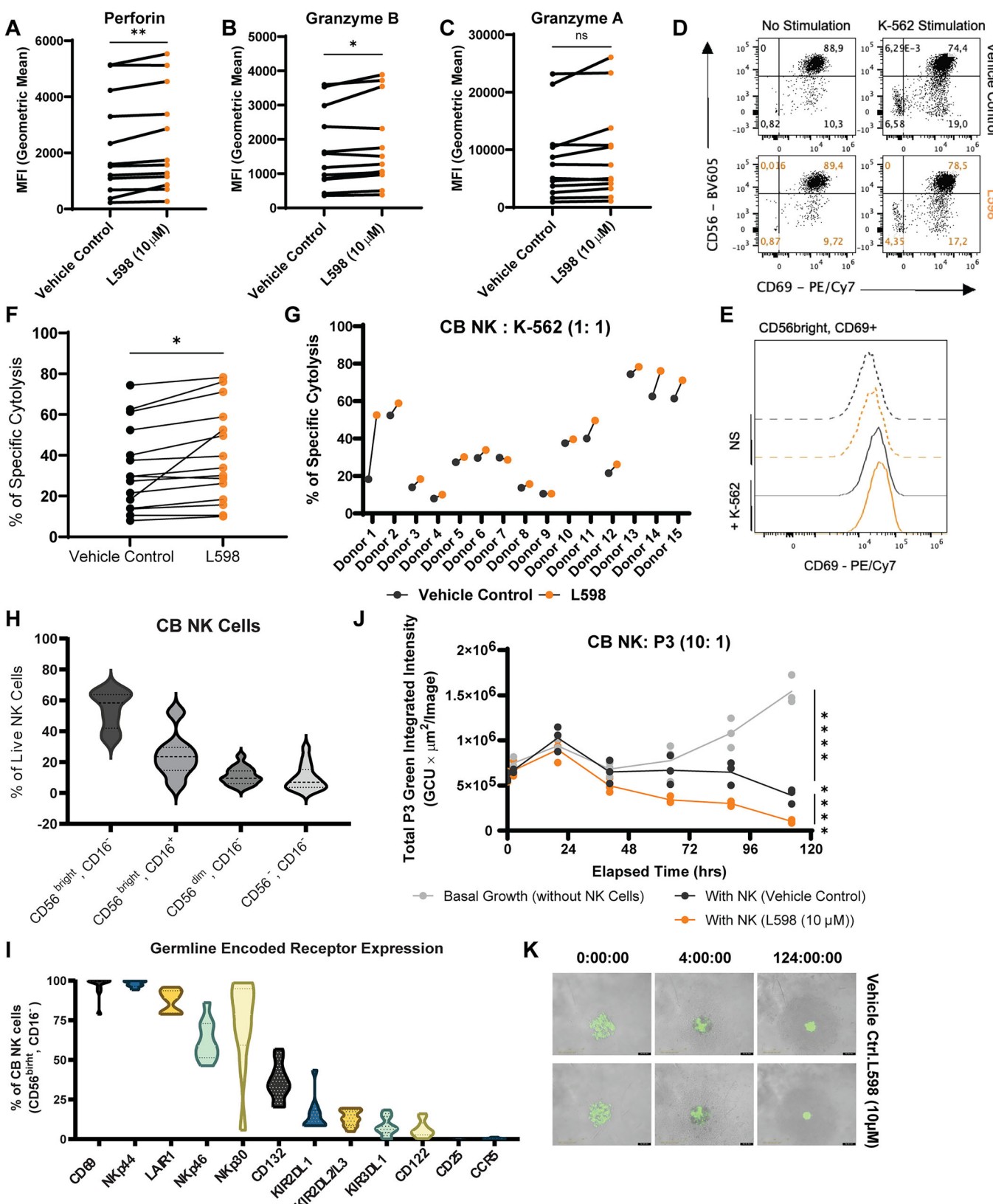

*C.-H. Feng et al*  EMBO reports

**Figure 7.  Applying dual inhibition of PTPN1 and PTPN2 to improve umbilical cord blood NK cell therapy.**

(A–C) Flow cytometry analysis of intracellular cytolytic granules expression (granzyme A, granzyme B, and perforin) in CB NK cells treated for 5 days with the dual inhibitor L598 (10 μM) or vehicle control (*n* = 12 donors, each tested in one or two or three technical replicates). Each dot represents the average value of technical replicates from a single donor. *P*-values: *p* = 0.0072 (**) for Perforin, *p* = 0.0199 (*) for Granzyme B, *p* = 0.0595 (ns) for Granzyme A (Paired t-test). (D, E) A slight increase in the percentage and mean fluorescence intensity (MFI) of CD69 was observed in live CD56⁺ cord blood NK cells treated with L598 (10 μM) for 6 days in culture, compared to vehicle controls. A representative figure of results from *n* = two different donors tested in duplicates is shown. (F, G) Flow cytometry-based anti-tumor cytolysis assay (E:T = 1:1) against K-562 tumor target cells by CB NK cells with L598 (10 μM or 30 μM) or vehicle control treatment for 6 days in cell culture. Pooled results from *n* = 15 donors in (F) and individual donor response in triplicates or duplicates or no-replicates in (G) are shown. Each dot represents the mean of technical replicates from a single donor. *P*-value: *p* = 0.0102 (*) (Paired t-test). (H) Expression of CD56 and CD16 on CB NK cells varied across the 15 donors. Among the identified subsets, CD56^bright^CD16⁻ and CD56^bright^CD16⁺ populations showed the greatest inter-donor variability. Dotted lines indicate the 25th, median (50th), and 75th percentile values. (I) Flow cytometry analysis of CD56^bright^, CD16⁻ CB NK cells for germ-line encoded surface receptor expression (*n* = 10 donors). Notable donor-to-donor differences were observed in NKp30, NKp46, CD132, and KIR2DL1 expression. Dotted lines indicate the 25th, median (50th), and 75th percentile values. (J, K) IncuCyte cytolysis assay of eGFP⁺ P3 cells targeted by CB NK cells from *n* = two potent donors (E:T = 10:1, in three technical replicates). Quantification of green fluorescence is shown in (J). *P*-values: p ≤ 0.0001 (****) (Two-way ANOVA with Tukey's multiple comparisons test). Source data are available online for this figure.

2018; Freud et al, 2017; Horowitz et al, 2013; Smith et al, 2020; Subedi et al, 2022; Yang et al, 2019). Although our receptor profiling was limited by conventional flow cytometry, we began assessing the expression of killer-cell immunoglobulin-like receptors (KIRs), which mediate recognition of polymorphic MHC class I molecules and display considerable haplotype diversity (Kelley et al, 2005; Pende et al, 2019). These were conducted along with other activating and inhibitory receptors that were expressed on CB NK cells (Fig. 7I). Our ongoing work aims to correlate functional and immunophenotypic diversity between PTPN1/N2 dual-inhibition-responsive and non-responsive CB NK donors. This broader receptor screen may also contribute to a more precise approach to CB NK cell-based anti-cancer immunotherapy.

In conclusion, our studies using the human NK-92 cell line and ex vivo-expanded human CB NK cells demonstrate that the homologous non-receptor protein tyrosine phosphatases PTPN1 and PTPN2 can be simultaneously targeted to enhance cytolytic activity against NK-sensitive tumor cells in vitro and in vivo. Collectively, these findings highlight a translational opportunity to apply dual targeting PTPN1 and PTPN2 as a strategy to improve the current NK cell-based anti-tumor immunotherapy for treating hard-to-treat cancers like glioblastoma and triple-negative breast cancers, within the immunosuppressive microenvironment.

## Methods

### Reagents and tools table

| Reagent/Resource | Reference or Source | Identifier or Catalog Number |
|---|---|---|
| **Experimental models** | | |
| NSG (NOD.Cg-Prkdcscid Il2rgtm1Wjl/SzJ) (*M. musculus*) | The Jackson Laboratory | 005557 |
| Cord Blood NK cells | Dr. Pierre Laneuville's and Dr. Linda Peltier's Lab (McGill University Health Center)/ This study | |
| NK-92 cells | ATCC | CRL-2407 |
| K-562 cells | ATCC | CCL-243 |

| Reagent/Resource | Reference or Source | Identifier or Catalog Number |
|---|---|---|
| Reh cells | ATCC | CRL-8286 |
| Jurkat cells | ATCC | TIB-152 |
| HEK293T/17 cells | ATCC | ACS-4500 |
| HCT116 cells | ATCC | CCL-247 |
| Caki-1 cells | ATCC | HTB-46 |
| U87 MG cells | ATCC | HTB-14 |
| THP-1 cells | ATCC | TIB-202 |
| MDA-MB-231 cells | ATCC | HTB-26 |
| MDA-MB-231 TR cells | Dr. Peter M. Siegel's Lab (McGill University) | (Tabariès et al, 2019) |
| P3 cells | Dr. Rolf Bjerkvig's Lab (University of Bergen) | (Daubon et al, 2019) |
| KG1A cells | Dr. François Mercier's Lab (McGill University)/ ATCC | (Fooks et al, 2022) |
| MOLM14 cells | Dr. François Mercier's Lab (McGill University) | (Fooks et al, 2022) |
| U937 cells | Dr. François Mercier's Lab (McGill University)/ ATCC | (Fooks et al, 2022) |
| **Recombinant DNA** | **Reference** | **Identifier** |
| pPrime-CMV-GFP-Mir30-PGK-Puro | Dr. Jerry Pelletier's Lab (McGill University) | (Robert et al, 2014) |
| **Flow Cytometry Antibodies** | **Source** | **Catalog Number** |
| CD56 – BV605 (Clone HCD56), 1:100 (NK-92), 1:50 (CB NK), and 1:200 (Tumor infiltrated cells) | BioLegend | 318334 |
| CD56 – APC (Clone HCD56), 1:200 (Tumor infiltrated cells) | BioLegend | 318310 |
| CD69 – PE/Cy7 (Clone FN50), 1:50 | BioLegend | 310912 |
| CD69 – BV421 (Clone FN50), 1:200 (Tumor infiltrated cells) | BioLegend | 310930 |

EMBO reports Volume 27 | May 2026 | 2581 – 2613  **2595**

| Reagent/Resource | Reference or Source | Identifier or Catalog Number |
| --- | --- | --- |
| CD25 – PE/Cy7 (Clone BC96), 1:50 | BioLegend | 302612 |
| CD122 – APC (Clone Tu27), 1:50 | BioLegend | 339008 |
| CD34 – BUV 737 (Clone 563), 1:100 | BD Bioscience | 741868 |
| CD45 – PE/Cy7 (Clone HI30), 1:100 (CB NK) and 1:200 (Tumor infiltrated cells) | BioLegend | 304016 |
| CD3 – APC/Cy7 (Clone HIT3a), 1:50 | BioLegend | 300318 |
| CD14 – Biotin (Clone HCD14), 1:100 | BioLegend | 325624 |
| CD33 – Biotin (Clone WM53), 1:100 | BioLegend | 303426 |
| CD19 – Biotin (Clone HIB19), 1:100 | BioLegend | 302204 |
| CD66b – Biotin (Clone G10F5), 1:100 | BioLegend | 305120 |
| Streptavidin – APC/Cy7, 1:100 | BioLegend | 405208 |
| CD16 – Alexa Fluor 700 (Clone 3G8), 1:50 | BioLegend | 302026 |
| CD16 – FITC (Clone 3G8), 1:50 | BioLegend | 302006 |
| Granzyme A – APC (Clone CB9), 1:40 | BioLegend | 507220 |
| Granzyme B – BV405 (Clone GB11), 1:40 | BD Bioscience | 561151 |
| Perforin – FITC (Clone B-D48), 1:40 | BioLegend | 353310 |
| CD107a – APC (Clone H4A3), 1:100 | BioLegend | 328620 |
| CD107a – PE (Clone H4A3), 1:100 | BioLegend | 328608 |
| IFN-γ– FITC (Clone 4S.B3), 1:40 | BD Bioscience | 552882 |
| T-BET – PE (Clone 4B10), 1:100 | BioLegend | 644809 |
| EOMES – Alexa Fluor 647 (Clone X4-83), 1:100 | BD Bioscience | 567168 |
| HLA-A, B, C – FITC (Clone W6/32), 1:100 | BioLegend | 311403 |
| HLA-DR, DP, DQ – PerCP-Cy5.5 (Clone Tü39), 1:100 | BioLegend | 361709 |
| CD11c – BV510 (Clone N418), 1:200 | BioLegend | 117337 |
| MHC II – Alexa Fluor 700 (Clone M5/114.15.2), 1:400 | Invitrogen | 56-5321-80 |
| Ly6C – APC/Cy7 (Clone AL21), 1:400 | BD Bioscience | 560596 |
| CD11b – BV711 (Clone M1/70), 1:500 | BioLegend | 101241 |
| F4/80 – PE/Cy5 (Clone BM8), 1:500 | BioLegend | 123111 |
| Ly6G – PE/Cy7 (Clone 1A8), 1:400 | BioLegend | 127617 |
| CD14 – PE (Clone Rm-C5-3), 1:200 | BD Bioscience | 553740 |
| NKp30 – APC (Clone P30-15), 1:200 (Tumor infiltrated cells) | BioLegend | 325209 |
| NKp46 – PE (Clone 9E2), 1:200 (Tumor infiltrated cells) | BioLegend | 331908 |
| NKG2D – PerCP/Cy5.5 (Clone 1D11), 1:200 (Tumor infiltrated cells) | BioLegend | 320817 |
| NKG2A – BV711 (Clone 131411), 1:200 (Tumor infiltrated cells) | BD Bioscience | 747919 |
| PD-1 – BV785 (Clone EH12.2H7), 1:200 (Tumor infiltrated cells) | BioLegend | 329929 |
| Fixable Viability Dye – Zombie Red, 1:1000 | BioLegend | 423109 |

| Reagent/Resource | Reference or Source | Identifier or Catalog Number |
| --- | --- | --- |
| Fixable Viability Dye – eFluor 780, 1:2000 | Invitrogen | 65-0865-14 |
| **Western Blot Antibodies** | **Source** | **Catalog Number** |
| STAT1 (Clone Polyclonal) – Rabbit, 1:1000 | Cell Signaling Technology | 9172 |
| p-STAT1 (Y701) (Clone D4A7) – Rabbit, 1:1000 | Cell Signaling Technology | 7649 |
| STAT2 (Clone EP1814Y) – Rabbit, 1:2000 | Abcam | ab134192 |
| p-STAT2 (Y690) (Clone Polyclonal) – Rabbit, 1:1000 | Abcam | ab53132 |
| STAT3 (Clone 79D7) – Rabbit, 1:2000 | Cell Signaling Technology | 4904 |
| p-STAT3 (Y705) (Clone Polyclonal) – Rabbit, 1:1000 | Cell Signaling Technology | 9131 |
| STAT4 (Clone C46B10) – Rabbit, 1:1000 | Cell Signaling Technology | 2653 |
| p-STAT4 (Y693) (Clone Polyclonal) – Rabbit, 1:1000 | Cell Signaling Technology | 5267 |
| STAT5 (Clone D2O6Y) – Rabbit, 1:2000 | Cell Signaling Technology | 94205 |
| p-STAT5 (Y694) (Clone Polyclonal) – Rabbit, 1:1000 | Cell Signaling Technology | 9351 |
| STAT6 (Clone D3H4) – Rabbit, 1:2000 | Cell Signaling Technology | 5397 |
| p-STAT6 (Y641) (Clone Polyclonal) – Rabbit, 1:1000 | Cell Signaling Technology | 9361 |
| AKT (Clone Polyclonal) – Rabbit, 1:1000 | Cell Signaling Technology | 9272 |
| p-AKT (S473) (Clone Polyclonal) – Rabbit, 1:1000 | Cell Signaling Technology | 9271 |
| p38 MAPK (Clone Polyclonal) – Rabbit, 1:1000 | Cell Signaling Technology | 9212 |
| p-p38 (T180/Y182) (Clone Polyclonal) – Rabbit, 1:1000 | Cell Signaling Technology | 9211 |
| p44/42 (Erk1/2) (Clone Polyclonal) – Rabbit, 1:1000 | Cell Signaling Technology | 9102 |
| p-p44/42 (Erk1/2) (T202/Y204) (Clone E10) – Mouse, 1:1000 | Cell Signaling Technology | 9106 |
| SHP-2 (Clone D50F2) – Rabbit, 1:1000 | Cell Signaling Technology | 3397 |
| p-SHP-2 (Y542) (Clone Polyclonal) – Rabbit, 1:1000 | Cell Signaling Technology | 3751 |
| SMAD2 (Clone L16D3) – Mouse, 1:1000 | Cell Signaling Technology | 3103 |
| p-SMAD2 (S465/S467) (Clone 138D4) – Rabbit, 1:1000 | Cell Signaling Technology | 3108 |
| SOCS 1 (Clone A156) – Rabbit, 1:1000 | Cell Signaling Technology | 3950 |
| SOCS 3 (Clone Polyclonal) – Rabbit, 1:1000 | Cell Signaling Technology | 2923 |
| Granzyme A (Clone EPR20161) – Rabbit, 1:1000 | Abcam | ab209205 |

| Reagent/Resource | Reference or Source | Identifier or Catalog Number |
|---|---|---|
| Granzyme B (Clone Polyclonal) – Rabbit, 1:2000 | Cell Signaling Technology | 4275 |
| Perforin (Clone B-D48) – Mouse, 1:1000 | Abcam | ab47225 |
| PTPN1 (Clone 15) – Mouse, 1:1000 | BD Transduction Laboratories | 610140 |
| PTPN2 (Clone 3E2) – Mouse, 1:2000 | Tremblay Lab Generation | / |
| Vinculin (Clone hVIN-1) – Mouse, 1:10,000 | Sigma-Aldrich | V9131 |
| Beta-Actin (Clone 13E5) – Rabbit, 1:2000 | Cell Signaling Technology | 4970 |
| **Oligonucleotides and other sequence-based reagents** | | |
| PPIB_F_ GCACAGGAGGAAAGAGCATC PPIB_R_ AGCCAGGCTGTCTTGACTGT | This study | / |
| Actin B_F_ CTCTTCCAGCCTTCCTTCCT Actin B_R_ AGCACTGTGTTGGCGTACAG | This study | / |
| PTPN1_F_ GGCCCGTCATGGAGATGGAA PTPN1_R_ GCTTGGCCACTCTACATGGGA | This study | / |
| PTPN2_F_ TCGAGCGGGAGTTCGAAGAG PTPN2_R_ CACGACTGTGATCATATGGGCT | This study | / |
| PTPN6_F_ TGGCGTGGCAGGAGAACAG PTPN6_R_ GCAGTTGGTCACAGAGTAGGGC | (Christophi et al, 2008) | / |
| PTPN11_F_ CACTACAGCGCAGGATTGAA PTPN11_R_ GGCTCTGATCTCCACTCGTC | This study | / |
| Granzyme A_F_ TCAGGTTGATTGATGTGGGACAG Granzyme A_R_ GACCATGTAGGGTCTTGAATGAGGA | (Zhang et al, 2008) | / |
| Granzyme B_F_ TGGGGGACCCAGAGATTAAAA Granzyme B_R_ TTTCGTCCATAGGAGACAATGC | (Chang et al, 2016) | / |
| Granzyme H_F_ TGGCGGCATCCTAGTGAGAA Granzyme H_R_ GCCCCCAAGGTGACATTTATG | This study | / |
| Granzyme M_F_ GGACACCCGCATGTGTAACAAC Granzyme M_R_ GATGTCAGTGCAGACCCTGGAG | This study | / |
| Granzyme K_F_ ATCAACACATTTCATCTGGGCTTC Granzyme K_R_ AAACGTGATGTCCGCCATACTG | This study | / |

| Reagent/Resource | Reference or Source | Identifier or Catalog Number |
|---|---|---|
| PRF 1_F_ CGCCTACCTCAGGCTTATCTC PRF 1_R_ CCTCGACAGTCAGGCAGTC | (Morissette et al, 2007) | / |
| IFNG_F_ AAGTGATGGCTGAACTGTCG IFNG_R_ GCAGGCAGGACAACCATTAC | This study | / |
| **Chemicals, Enzymes and other reagents** | | |
| L598 | (Montalibet et al, 2006) | / |
| KQ-791 | (Therien et al, 2013) | / |
| PMA (Phorbol 12-Myristate 13-Acetate) | Millipore Sigma | 5005820001 |
| Ionomycin | Sigma-Aldrich | 407953 |
| rhIL-2 | PeproTech | 200-02 |
| rhIL-4 | PeproTech | 200-04 |
| rhIL-7 | STEMCELL Technologies | 78053.1 |
| rhIL-12 p70 | PeproTech | 200-12 |
| rhIL-15 | STEMCELL Technologies | 78218 |
| rhIL-35 | PeproTech | 200-37 |
| rhFlt3/Flk-2 | STEMCELL Technologies | 78009 |
| rhSCF | STEMCELL Technologies | 78062 |
| rhTGF-β1 | PeproTech | 100-21 |
| rhTGF-β RII | Biotechne/R&D Systems | 241-R2 |
| **Software** | | |
| Prism 9.4.1 | https://www.graphpad.com/updates/prism-941-release-notes | / |
| FlowJo 10.8.1 | https://www.flowjo.com/flowjo10/overview | / |
| **Other** | | |

## Human cell lines and primary NK cell culture

Human NK cell line NK-92 (CRL-2407) and other human cell lines: K-562 (CCL-243), Reh (CRL-8286), Jurkat (TIB-152), HEK293T/17 (ACS-4500), HCT116 (CCL-247), Caki-1 (HTB-46), U87 MG (HTB-14) and THP-1 (TIB-202) were originally from American Type Culture Collection (ATCC) and cultured according to the manufacturer's protocols. Patient-derived primary glioblastoma P3 cells were kindly provided by Dr. Rolf Bjerkvig (University of Bergen) and were cultured in neurobasal medium (Thermo Fisher Scientific, Cat. 21103049) supplemented with B-27 supplement, 1%

of Penicillin/ Streptomycin, 10 ng/mL basic Fibroblast Growth Factor (FGFb), and 10 U/mL Heparin. P3 cells expressing eGFP were generated by lentivirus infection with a pLenti6-eGFP vector. MDA-MB-231 (HTB-26, ATCC) and MDA-MB-231 TR cell lines were kindly gifted from Dr. Peter M. Siegel (Rosalind and Morris Goodman Cancer Institute - GCI, McGill University) and were cultured in DMEM- High glucose (HyClone, Cat. SH3002201) media supplemented with 10% of heat-inactivated FBS and 1% of Penicillin/ Streptomycin. AML cell lines: KG1A, MOLM14, and U937 were kindly gifted from Dr. François Mercier (Lady Davis Institute for Medical Research, McGill University). KG1A cells were cultured in Iscove's Modified Dulbecco's Medium (WISENT, Cat. 319-105 CL) with 20% heat-inactivated FBS and 1% Penicillin/ Streptomycin. U937, and MOLM-14 cells were cultured in RPMI 1640 (HyClone, Cat. SH3002701), supplemented with 10% heat-inactivated FBS and 1% Penicillin/Streptomycin. All cell lines were maintained in 5% $CO_2$, 37 °C incubator, except that NK-92 cells were maintained in 6.5% $CO_2$ incubator at 37 °C. All cell lines were tested regularly for mycoplasma infection using PCR and were confirmed as negative. All cell lines except those gifted from other labs have been authenticated by STR profiling service.

Human CB NK cells from both sexes were kindly provided by Dr. Linda Peltier (Cellular Therapy Laboratory at the Research Institute of McGill University Health Center in Montreal, Canada) in frozen format and were stored in a liquid nitrogen tank until usage. Briefly, CB NK cells were negatively isolated from the buffy coat of collected cord blood units with EasySep™ Human NK cell Isolation Kit (StemCell Technology, Cat. 17955). Isolated NK cells were ex vivo expanded for $14 \pm 2$ days in ImmunoCult™-XF T Cell Expansion Medium (StemCell Technology, Cat. 10981) supplemented with 10% cord plasma and with a cytokine cocktail containing IL-2, IL-7, IL-15, FLT3/L, and SCF. Expanded NK cells were cryopreserved in cytokine-free, complete media supplemented with 10% Dextran40/DMSO and were shipped to the Goodman Cancer Institute (Montreal, Canada). For experiments, CB NK cells were thawed and rested for 2 to 3 days in ImmunoCult™-XF T Cell Expansion Medium supplemented with 10% human male AB Serum (Sigma-Aldrich, Cat. H4522), 1% of Penicillin/ Streptomycin, 1× GlutaMax (Thermo Fisher Scientific, Cat. 35050061) and a cytokine cocktail containing IL-2 (5 ng/mL), IL-7 (5 ng/mL), IL-15 (5 ng/mL), SCF (15 ng/mL) and FLT3/L (10 ng/mL). After resting, CB NK cells were plated in complete media with the additions of PTPN1/PTPN2 dual-inhibitor (10 µM or 30 µM) or the vehicle control, which were renewed with the cytokine cocktail every 3 days. Intracellular cytolytic granules examination and anti-tumor cytolysis assays were performed following 5 days or 6 days of cell culture, respectively. For anti-tumor cytolysis assays, IL-2 concentration was increased to 25 ng/mL during cell culture. RNA extraction from CB NK cells was done following 3 days of cell culture without the dual-inhibitor or vehicle control treatment.

## Lentiviral transduction

PTPN1 and/or PTPN2 deficient cells: sh*PTPN1*, sh*PTPN2*, sh*PTPN1/N2* (dKD) and the control shFF cells were generated by lentivirus spinoculation according to the previously published protocol (Sutlu et al, 2012). Briefly, one out of four constructs of sh*PTPN1* or sh*PTPN2* in the pPRIME-CMV-GFP-Mir-PGK-Puro vector was selected to transfect HEK293/T17 cells with

Lipofectamine 2000 (Thermo Fisher Scientific, Cat. 11668019) according to the manufacturer's protocol; cell supernatant was collected 24 h and 48 h post-transfection and was concentrated with Viro-PEG Lentivirus Concentrator (OZ Bioscience, Cat. LVG100). Concentrated lentivirus particles were resuspended in virus storage buffer (20 mM PIPES, 75 mM NaCl, 2.5% sucrose, pH at 6.5) and stored at −80 °C freezer until usage (Deb et al, 2017). Lentiviral tittering was performed with HCT116 cells and assessed by the percentage of viable GFP+ cells using flow cytometry. Spinoculation of NK-92 cells was performed in a RetroNectin-coated 24-well plate format. NK-92 cells were stimulated with cytokines IL-2 (1000 IU/mL) and IL-12 (100 ng/mL) for 2 h before spinoculation. BX795 (working concentration at 6 µM) was added to NK-92 cells to enhance lentivirus spinoculation efficiency. Lentiviral particles were added at M.O.I. = 30 to NK-92 cells per well. Viable and GFPhigh transduced cells were sorted by flow cytometry at the Flow Cytometry Core Facility, McGill University and were expanded in regular NK-92 cell culture conditions. sh*PTPN1/N2* dKD cells were generated by serial knockdowns of PTPN1 and PTPN2.

## Quantitative real-time RT-PCR

Total RNA was isolated with Qiagen RNeasy Mini Kit without DNase I on-column digestion, according to the manufacturer's protocol. Purified total RNA was resuspended in DNase-free, RNase-free, and Protease-free sterile water and quantified with NanoDrop. RNA integrity was checked by gel electrophoresis. RNA (500 ng) was reverse transcribed into cDNA with the SuperScript III Reverse Transcriptase Kit (Thermo Fisher Scientific, Cat. 18080085), using random hexamers and Oligo(dT) 20 primer and followed by RNaseH treatment, according to the manufacturer's protocol. qRT-PCR was performed using FastStart Essential DNA Green Master (Roche Diagnostics, Cat. 06924204001) according to the manufacturer's instructions and a Bio-Rad CFX96 Real-Time PCR Detection System. Pfaffl method with reference gene *PPIB* was used for analysis of *PTPN1* and *PTPN2* in NK-92 and Jurkat cells. Vandesompele method with reference genes *PPIB* and *ACTB* was used for analysis in CB NK cells. ΔΔCt method with reference genes: *PPIB* (*GZMB, IFNG, GZMA* and *GZMK*) and *ACTB* (*PRF1, GZMH* and *GZMM*) was used for analysis of cytolytic effector molecules in NK-92 cells. Primer sequence information is shown in Reagents and Tools Table, including references for primers (Chang et al, 2016; Christophi et al, 2008; Morissette et al, 2007; Zhang et al, 2008).

## Enzyme-linked immunosorbent assay (ELISA)

Cell supernatant was collected according to the following procedures: NK-92 cells pre-treated in 30 µM L598, or vehicle control were collected, counted, and starved in serum-free media for 30 min. After starvation, NK-92 cells were counted and plated at 600,000 cells per well in serum-free media, followed by the addition of K-562 cells at an effector to target cell ratio (E:T) at 6:1 for the tumor target cell stimulation group, or by the addition of serum-free K-562 media for the no stimulation group. To remove background cytokine secretion by K-562 cells, an equal amount (100,000 cells) of K-562 cells was plated in serum-free, mixed K-562 and NK-92 cell culture media. Cells were incubated in a

37 °C, 5% $CO_2$ incubator for 4 h and the supernatant was collected after centrifugation. The supernatant was frozen at −20 °C until the time of the assay. Secreted human IFN-γ concentration was analyzed with the Human IFN-γ ELISA Kit (R&D, DIF50) according to the manufacturer's protocol. Results were normalized against IFN-γ secreted by K-562 cell-only condition.

## Flow cytometry-based tumor cytolysis assays

Tumor target cells K-562 and Reh were labeled with cell tracker CFSE (2.5 μM) or CMRA (5 μM) (Invitrogen, Cat. C34551) 1 day before the cytolysis experiment, according to the manufacturer's protocol. Labeled tumor target cells were washed once with PBS and resuspended in complete cell media. NK cells were washed and resuspended in complete media without cytokines. NK cells and tumor target cells were plated at various E:T ratios with tumor target cells plated consistently at 50,000 cells per well in a round-bottom 96-well plate. The mixed cells were co-cultured in a 5% $CO_2$, 37 °C incubator, for 3 h (NK-92 cytolysis assays) or for 5 h (shRNA modified NK-92 and CB NK cytolysis assay). For the TGF-β blockade and NK-92 cytolysis assay, soluble sTβRII (5 nM) was used to neutralize TGF-β1 (1 ng/mL), with or without L598 (10 μM), for 45 min prior to a 3-day treatment of NK-92 cells. For CB NK cell cytolysis assay, K-562 cells were plated to NK cells at E:T (1:1) and the plate was centrifuged at $100 \times g$ for 1 min at room temperature before co-culture. After co-culture, cells were washed once with ice-cold PBS and then stained with fixable viability dyes: Aqua (Invitrogen) or UV Blue (Invitrogen) or eFluor 780 (eBioscience) at 1:2000 in dark for 20 min at 4 °C. Stained cells were washed twice with ice-cold PBS and were fixed with 1% Paraformaldehyde (PFA) for 1 h or 2% PFA for 30 min in dark at 4 °C. Flow cytometry sample acquisition was performed with BD LSRFortessa and data was analyzed with FlowJo 10.8.1. The calculation for % of specific cytolysis was followed by the equation:

$$\% \text{ of Specific Cytolysis} = \frac{100 \times (\% \text{ of Dead Target Cells} - \% \text{ of Basal Death of Target Cells})}{100 - \% \text{ of Basal Death of Target Cells}}$$

## Surface and intracellular flow cytometry

For surface protein assessments, around 200,000 cells were collected and washed once with ice-cold PBS, followed by viability staining with fixable viability dye eFluor780 for 20 min on ice. After viability staining, cells were washed twice with ice-cold FACS buffer (2% BSA/PBS) followed by Fc receptors blocking with Human TruStain FcX (1:25, BioLegend Cat. 422302) at room temperature for 10 min, which was removed after incubation. Cells were then stained for 30 min on ice in antibody cocktails prepared in FACS buffer. After antibody cocktail staining, cells were washed twice with ice-cold FACS buffer and then fixed with 2% PFA on ice for 20 min, followed by cell washes and resuspension in FACS buffer. For CB NK cell staining, an additional step using APC/Cyanine7 Streptavidin was performed on ice for 30 min before cell fixation. The list of antibody cocktails for NK-92, CB NK cells and tumor target cells, respectively, and antibody dilutions were presented in the Reagents and Tools Table.

For intracellular cytolytic granule staining, around 200,000 cells (NK-92) or 100,000 cells (CB NK) were collected and washed once with ice-cold PBS and stained with fixable viability dye as mentioned previously. After viability staining, cells were washed twice with ice-cold PBS and fixed with BD Cytofix/Cytoperm solution (BD Biosciences, Cat. 554714) on ice for 20 min, followed by washing twice with ice-cold Perm/Wash buffer. After perm washes, cells were blocked with Human TruStain FcX prepared in Perm/Wash buffer, as mentioned previously. After Fc Receptor blocking, cells were stained on ice for 30 min with antibodies targeting granzyme A (APC, BioLegend), granzyme B (BV450, BD Bioscience), and perforin (FITC, BioLegend), which was prepared in Perm/Wash buffer. After intracellular staining, cells were washed twice with Perm/Wash buffer and resuspended in 2% BSA/PBS buffer. All incubations were carried in the dark. Samples data were acquired with BD LSR Fortessa and results were analyzed by FlowJo 10.8.1. The gating strategies are shown in Appendix Figs. S4 and S5, according to the MIFlowCyt guideline.

## Degranulation and cytokine production assay

NK-92 cells pre-treated with L598 (30 μM) or vehicle control were stimulated with either K-562 (E:T at 1:1) or PMA/Ionomycin (1 μg/mL each) or with K-562 complete media (no-stimulation control), respectively. Anti-CD107a or LAMP-1 antibody was added to cells before incubation. Golgi stop (containing Monensin, 1 μM, BD Bioscience Cat. 554724) and Golgi plug (containing Brefeldin A, 4 μg/mL, BD Bioscience Cat. 555029) were added to cells 1 h after the degranulation assay started and were kept until the end of the 6-h-degranulation assay. For the degranulation kinetic assay, NK-92 cells pre-treated with L598 (10 μM) or vehicle control were co-cultured with K-562 (E:T at 1:1) for a time-course, ranging from 15 min to 60 min, without the addition of Golgi stop and Golgi plug treatment. Intracellular staining of interferon-γ (FITC, BD Bioscience) was performed as described, except that Fc Receptor Blocking was performed for 5 min at room temperature in the dark. Cell gating and the percentage of population quantification were based on no stimulation control. Flow cytometry samples were acquired with BD LSR Fortessa. The gating strategy is shown in Appendix Fig. S5, according to the MIFlowCyt guideline. Flow cytometry antibody information and dilution are shown in the Reagents and Tools Table.

The rates for upregulation of CD107a and degranulation of granzyme A, granzyme B and perforin was calculated based on no stimulation controls at each time points, based on the following formula:

Rate of CD107a Upregulation
$$= \frac{\text{CD107a MFI (Tumour Stimulation for X } mins) - \text{CD107a MFI (No Stimulation for X } mins)}{\text{Elapsed Time X } (mins)}$$

*Or*

Rate of Degranulation
$$= \frac{\text{MFI (No Stimulation for X } mins) - \text{MFI (Tumour Stimulation for X } mins)}{\text{Elapsed Time X } (mins)}$$

## Immunoblotting and cytokine stimulation assay

Total cell lysates were prepared using modified RIPA buffer (1 M Tris-HCl pH 7.5, 5 M NaCl, 0.25% sodium deoxycholate, 10% NP-

40, 10 mM NaF), supplemented with EDTA-free protease inhibitor (Sigma-Aldrich, Cat. 11873580001) and 2 mM $Na_3VO_4$. Protein concentration was measured using the bicinchoninic acid (BCA) assay, and equal amounts were resolved on 10% SDS-PAGE gels containing 2,2,2-Trichloroethanol (Sigma-Aldrich, Cat. T54801-100G) for total protein visualization. Proteins were transferred to PVDF membranes (Sigma-Aldrich, Cat. IPVH00005) using the Trans-Blot Turbo Transfer System (Bio-Rad), blocked with 5% BSA in PBS-T (0.1% Tween-20), and incubated with primary antibodies overnight at 4 °C. After washing, membranes were incubated with HRP-conjugated secondary antibodies (1:10,000, Jackson Immuno-noResearch) for 1 h at room temperature, followed by enhanced chemiluminescence (ECL) detection using ChemiDoc XRS+ (Bio-Rad).

For cytokine stimulation, NK-92 cells pre-treated with dual inhibitor L598 (30 µM) or vehicle control, or from different genetic modification backgrounds were serum-starved for 6 h. Cells were then stimulated with IL-2, or IL-15 at 37 °C. Stimulation duration and concentrations were indicated in figure legends. Stimulation was stopped by adding an equal volume of 2× TNE buffer (with 2× protease inhibitors, 10 mM NaF, 4 mM $Na_3VO_4$, and 20 mM $Na_4P_2O_7$), followed by boiling in Laemmli buffer to prepare lysates. Western blot analysis was performed using ImageLab software (version 6.1). Antibody details and dilutions are listed in the Reagents and Tools Table.

## IncuCyte-based cytolysis assay

For cytolysis assays using Caki-1, U87, and MDA-MB-231 cells, eGFP-labeled tumor cells were plated in 96-well flat-bottom plates at 10,000 cells/well (Caki-1, MDA-MB-231) or 15,000 cells/well (U87), and incubated overnight at 37 °C with 5% $CO_2$. For the P3 spheroid assay, eGFP-labeled P3 cells were dissociated with Accutase (Corning, Cat. 25058CI) and counted on the experiment day. Spheroids were formed by plating 1000 cells/well in Ultra-Low Attachment 96-well plates using complete neurobasal medium supplemented with 0.4% methylcellulose and IncuCyte Caspase-3/7 Red dye (1:400; Sartorius, Cat. 4704). After 4–6 h of incubation at 37 °C with 5% $CO_2$, spheroid formation and cell viability were assessed by IncuCyte S3 live imaging. NK cells were washed once with cytokine-free media and resuspended in fresh media supplemented with IL-15 (2 ng/mL) for CB NK assays. NK cells were then added to target cells at effector-to-target (E:T) ratios specified in the figure legends. Live-cell imaging was performed for up to 5 days. For tumor-only controls, NK cell medium (with matching cytokines) was added without NK cells. Cytolysis was analyzed using Top-hat segmentation with Threshold (GCU) values of 2.0 for P3 assays and 10.0 for Caki-1, U87, and MDA-MB-231 assays.

## Xenograft NSG-U87 model

Two cohorts of male NSG mice (The Jackson Laboratory, Cat. 005557), aged 11–14 weeks (for sh*PTPN1/PTPN2* dKD NK therapy, $n = 13$) or 8 weeks (for pharmacological dual inhibition of PTPN1/PTPN2 NK therapy, $n = 45$) were randomly and blindly assigned for dual-inhibitor treated NK therapy or control NK therapy or no treatment, according to ARRIVE. Tumor size was measured three times for each mouse, and the average value was reported. Mice

were housed at room temperatures of 18–24 °C and relative humidity of 30–70%. Lighting was maintained on a regular diurnal cycle with a photoperiod of 12–14 h of light. Food and water were provided ad libitum. For dual inhibition PTPN1/PTPN2 NK therapy assay, mice were produced by in vitro fertilization from the same male parental mouse to reduce genetic variability. Human glioblastoma cell line U87 was prepared in PBS (for dKD NK therapy) or mixed with an equal volume of Matrigel (for dual inhibition NK therapy) (Corning, Cat. 354262) and transplanted to NSG mice subcutaneously at 3 million cells per mouse. NK cell therapy started when the tumor size reached at least 50 mm³ measured using a digital caliper, at a dose of 3 millions cells per injection and once per week frequency for a total of 3 consecutive weeks (for dKD NK therapy) OR at a dose of 5 millions cells per injection and twice per week frequency for a total of 4.5 weeks (for dual inhibition NK therapy). In the dual inhibition NK therapy experiment, on day 18, mice died accidentally ($n = 5$ from the dual-inhibitor treatment group; $n = 3$ from the vehicle control group; $n = 2$ from the no treatment group) during i.v. injections possibly due to overheating. The tumor sizes were measured every 2–4 days for comparisons among the sh*FF* NK-92 treatment group ($n = 3$), the sh*PTPN1/N2* NK-92 treatment group ($n = 3$), and the no treatment group ($n = 7$) or among the sh*FF* with L598 treatment group ($n = 5$), the sh*FF* with vehicle control treatment group ($n = 7$) and the no treatment group ($n = 8$). For the dual inhibition NK therapy experiment, the mouse's body weight was measured every 2 days and photos of the tumor shape were taken every 7 days. Mice were sacrificed when the biggest tumor volume reached 2000 mm³ or showed abnormal body conditions according to SOP# 415 Humane Intervention Points for Rodent Cancer Models, McGill University and Canadian Council on Animal Care guidelines. At the humane intervention point, blood, tumors, and organs (heart, liver, lung, kidney, spleen, and lymph nodes) were collected for histological and flow cytometry analysis.

## Xenograft NSG-MDA-MB-231TR model

Two cohorts of female NSG mice (8–12 weeks old) were used for evaluating primary tumor growth ($n = 15$) and metastasis progression ($n = 23$). Mice were housed under the same conditions as described for the NSG-U87 model. Mice were orthotopically inoculated with 0.25 million MDA-MB-231TR cells in a 1:1 mixture of PBS and Matrigel into the mammary fat pad. Preoperative analgesia (Carprofen, 20 mg/kg) and local anesthesia (lidocaine/bupivacaine) were administered prior to xenograft implantation. Postoperative care included daily administration of Carprofen for 3 days. Tumor growth was monitored every 2 or 3 days using a digital caliper. When tumors reached a volume of 30–50 mm³, mice were randomized into three treatment groups: (1) NK-92 cells pre-treated with KQ791 (10 µM), (2) NK-92 cells with vehicle control (PBS), and (3) HBSS (cell-free control). NK-92 cells were administrated intravenously twice per week at 5 million cells per dose, in HBSS for 4 weeks. For the primary tumor growth control assay, mice were euthanized when the largest tumor reached 2000 mm³ or when animals exhibited signs of distress in accordance with SOP# 415 Humane Intervention Points. For the metastasis control assay, primary tumors were surgically resected when the average tumor size in the smallest group reached approximately 500 mm³. Resected tumors were weighed and

dissociated for flow cytometry analysis. Following surgery, mice were allowed to recover before being imaged using IVIS to detect initial metastatic signals. Metastasis progression was monitored by IVIS every 2 to 3 days. Mice were euthanized when any individual mouse showed abnormal body conditions, such as paralyzed limbs, according to SOP# 415. At the humane intervention point, blood, tumors, and organs (lung, liver, spleen, and lymph nodes) were collected for histological analysis, flow cytometry, and metastasis assessment. All procedures were conducted in compliance with McGill's standard operating procedures for anesthesia, analgesia, and euthanasia.

## Statistical analysis

As indicated in the figure legends, statistical analyses were performed in Prism 9.4.1 (GraphPad). Unpaired t-test (parametric), paired t-tests (parametric), One-Way ANOVA (Dunnett's test or Tukey's test), and Two-Way ANOVA (Holm-Šídák's test or Tukey's test) were selected according to the number and type of predictor variables. $P$-values: $p > 0.1234$ was denoted as not significant (ns); $p \leq 0.0332$ was denoted as *; $p \leq 0.0021$ was denoted as **; $p \leq 0.0002$ was denoted as ***; $p \leq 0.0001$ was denoted as ****.

## Study approval

All in vivo experimental procedures in NSG mice were approved by McGill University's Animal Care and Use Committee and were performed in accordance with the Canadian Council of Animal Care guidelines. Written informed consents were received from umbilical cord blood donors. Umbilical cord blood sample collection and processing were approved by McGill University Health Centre Ethical Committee and were performed in accordance with the Declaration of Helsinki and the Belmont Report.

# Data availability

This study includes no data deposited in external repositories.

The source data of this paper are collected in the following database record: biostudies:S-SCDT-10_1038-S44319-026-00745-0.

# Peer review information

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

## Acknowledgements

This work was supported by the Canadian Institute of Health Research Foundation grant (CIHR FDN-159923), Genome Canada and Génome Québec (Project PT 103179) through the Genomic Applications Partnership Program (GAPP) to MLT, the Richard and Edith Strauss Foundation and Fondation du cancer des Cèdres to PL. MLT holds a Chair of the Jeanne and Jean-Louis Levesque Foundation in Cancer Research. CHF is a recipient of Fonds de Recherche du Québec (FRQS) and acknowledged the financial support from the Department of Microbiology and Immunology Graduate Award at McGill University. We thank all cord blood donors and the support from the Cellular Therapy Laboratory at the McGill University Health Center. We thank Dr. Noriko Uetani, Dr. Claudia A. Penafuerte Diaz, Dr. Luis-Alberto Perez-Quintero and Dr. Serge Hardy for scientific discussions and reviewing this paper, and Dr. Janette Boudreau and Dr. Daniela Quail for commenting on this project. We are

grateful to Camille Stegen and Julien Leconte from the Flow Cytometry Core Facility (FCCF) at GCI, McGill University for cell sorting, and to Catherine Gagné and Karen H. Stone from the Comparative Medicine and Animal Resources Center (CMARC) at McGill University for assistance in intravenous injections. Our acknowledgment extends to Majid Ghahremani for his research support at GCI. We would like to recognize the support of Kanyr pharma Inc. for their kind gift of the KQ791 compound, Dr. Jerry Pelletier for providing miR-based shRNA constructs, Bianca Colalillo for the Jurkat cells, Yevgen Zolotarov and Dr. Arnim Pause for contributing the Caki-1 cells, Aaron Tachau for the human PTPN11 primers. Finally, this work was supported by the financial contribution of the McGill University Health Centre Foundation.

## Author contributions

**Chu-Han Feng**: Conceptualization; Data curation; Formal analysis; Validation; Investigation; Visualization; Methodology; Writing—original draft; Project administration; Writing—review and editing. **Linda Peltier**: Formal analysis; Investigation; Methodology; Writing—review and editing. **Tiffanie Chouleur**: Formal analysis; Investigation; Methodology; Writing—review and editing. **Milea DiPonzio**: Formal analysis; Validation; Investigation; Visualization. **Isabelle Aubry**: Formal analysis; Validation; Investigation. **Alexandre Poirier**: Formal analysis; Validation; Investigation; Writing—review and editing. **Zuzet M Cordova**: Validation; Investigation; Writing—review and editing. **Yunyun Shen**: Investigation; Methodology. **Sébastien Tabariès**: Methodology. **Xiaona Cao**: Methodology. **Guojun Chen**: Supervision; Methodology. **Andreas Bikfalvi**: Supervision; Methodology. **Silvia M Vidal**: Resources; Supervision; Methodology; Writing—review and editing. **Peter M Siegel**: Supervision; Methodology; Writing—review and editing. **Pierre Laneuville**: Conceptualization; Resources; Supervision; Funding acquisition; Methodology; Writing—review and editing. **Michel L Tremblay**: Conceptualization; Resources; Supervision; Funding acquisition; Methodology; Project administration; Writing—review and editing.

Source data underlying figure panels in this paper may have individual authorship assigned. Where available, figure panel/source data authorship is listed in the following database record: biostudies:S-SCDT-10_1038-S44319-026-00745-0.

## Disclosure and competing interests statement

Michel L. Tremblay is a founder of Kanyr Pharma Inc., a company developing small-molecule inhibitors of protein tyrosine phosphatases.

# Expanded View Figures

**Figure EV1. Genetic knockdown of PTPN1 and PTPN2 promotes NK-92 cell activation.**

(A, B) qRT-PCR analysis of PTPN1 and/or PTPN2 in shRNA-modified NK-92 cells. Results are from $n =$ three biological replicates. Data are presented as mean ± SD. *P*-values (*PTPN1*) in (A): $p = 0.0025$ (**) for sh*PTPN1/PTPN2* vs sh*FF*, $p \leq 0.0001$ (****) for sh*PTPN1* vs sh*FF* (One-way ANOVA with Dunnett's test). *P*-values (*PTPN2*) in (B): $p = 0.0014$ (**) for sh*PTPN1/PTPN2* vs sh*FF*, $p \leq 0.0001$ (****) for sh*PTPN2* vs sh*FF* (One-way ANOVA with Dunnett's test). (C) Flow cytometry analysis of cell viability in sh*FF*, sh*PTPN1*, sh*PTPN2*, and *shPTPN1/PTPN2* cells in regular cell culture conditions. Results are from $n =$ three biological replicates; each tested in three technical replicates). *P*-values: $p > 0.1234$ (ns) (One-way ANOVA with Tukey's test). (D–F) Cell images acquired in the bright phase using IncuCyte S3 were analyzed by ImageJ to quantify cell area for sh*FF* ($n = 707$ cells), sh*PTPN1* ($n = 700$ cells), sh*PTPN2* ($n = 594$ cells), and sh*PTPN1/PTPN2* ($n = 742$ cells) cells. Relative frequency in the percentage of cell area (bin size $= 20$) is presented in histography. *P*-values are indicated in the graph: $p < 0.0001$ (****) in (D); $p = 0.8277$ (ns) in (E); $p < 0.0001$ (****) in (F) (Unpaired t test). (G–J) Flow cytometry analyses of CD69 and CD25 expression on sh*FF* (control), sh*PTPN1*, sh*PTPN2* and sh*PTPN1/*sh*PTPN2* (dKD) cells following 5-h stimulation with tumor-target K562 (E:T $= 1:2$) or no stimulation. Geometric MFI and percentages of CD69$^+$ and CD25$^+$ live cells are shown in (G–J), with geoMFI in (G, I) and percentages in (H, J). Results are from $n =$ three biological replicates; each tested in three technical replicates. *P*-values in (G): $p = 0.0002$ (***) for sh*FF* vs sh*PTPN1* at no stimulation, $p = 0.0004$ (***) for sh*FF* vs sh*PTPN2* at no stimulation, $p = 0.0024$ (**) for sh*FF* vs sh*PTPN1* at E:T (1:2), $p = 0.0022$ (**) for sh*FF* vs sh*PTPN2* at E:T (1:2), $p = 0.0001$ (***) for sh*FF* vs sh*PTPN1/*sh*PTPN2* at E:T (1:2), $p \leq 0.0001$ (****). *P*-values in (H): $p = 0.0001$ (***) for sh*FF* vs sh*PTPN2* at no stimulation, $p = 0.0626$ (ns), $p \leq 0.0001$ (****). *P*-values in (I): $p = 0.0434$ (*) for sh*FF* vs sh*PTPN1* at no stimulation, $p = 0.9986$ (ns) for sh*FF* vs sh*PTPN2* at no stimulation, $p = 0.0001$ (***) for sh*FF* vs sh*PTPN1* at E:T (1:2), $p = 0.9653$ (ns) for sh*FF* vs sh*PTPN2* at E:T (1:2), $p \leq 0.0001$ (****). *P*-values in (J): $p = 0.3513$ (ns) for sh*FF* vs sh*PTPN1* at no stimulation, $p = 0.5773$ (ns) for sh*FF* vs sh*PTPN2* at no stimulation, $p = 0.0004$ (***) for sh*PTPN1* vs sh*PTPN1/*sh*PTPN2* at no stimulation, $p = 0.0002$ (***) for sh*FF* vs sh*PTPN1* at E:T (1:2), $p = 0.9456$ (ns) for sh*FF* vs sh*PTPN2* at E:T (1:2), $p \leq 0.0001$ (****). (Two-way ANOVA with Šídák's multiple comparisons test).

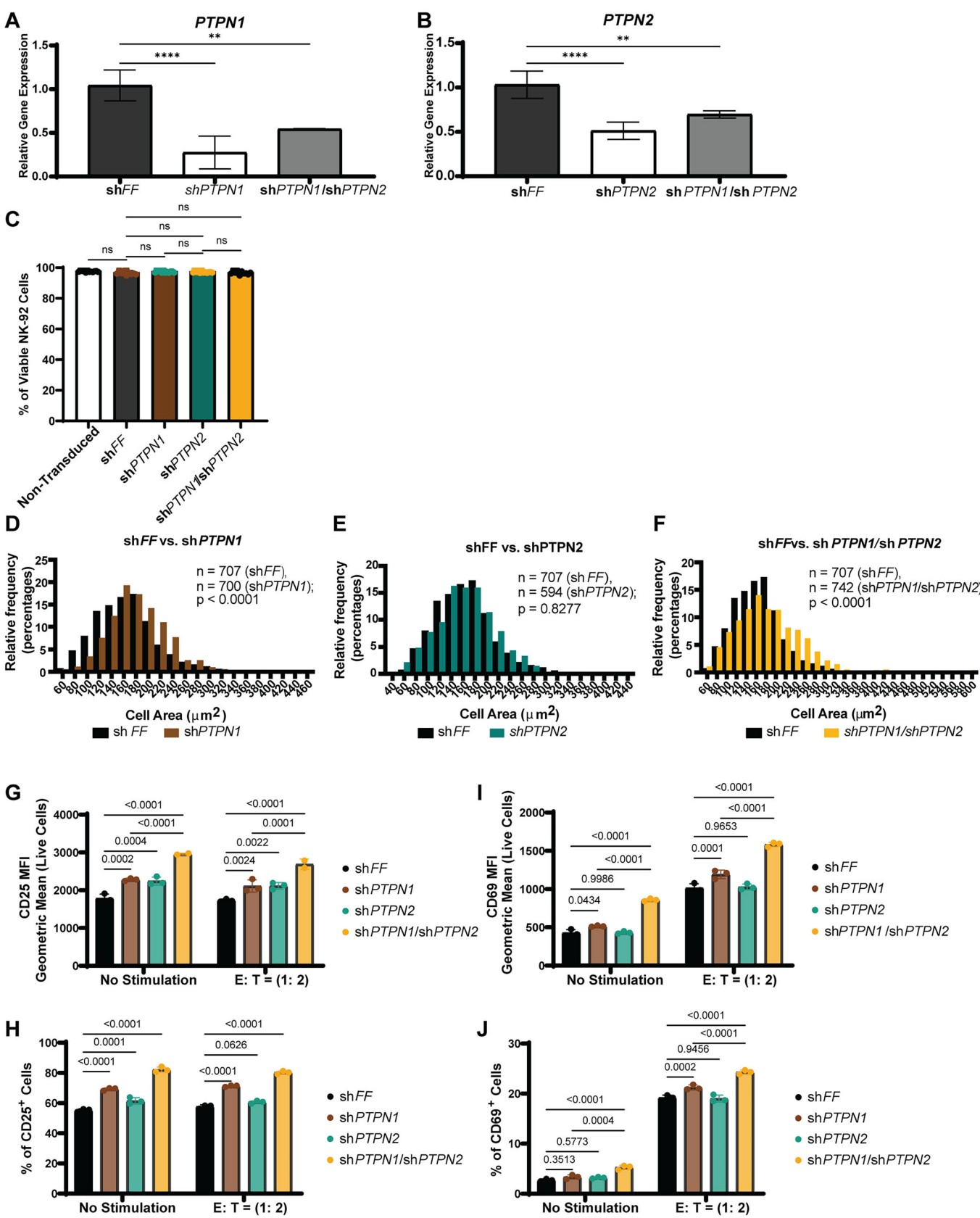

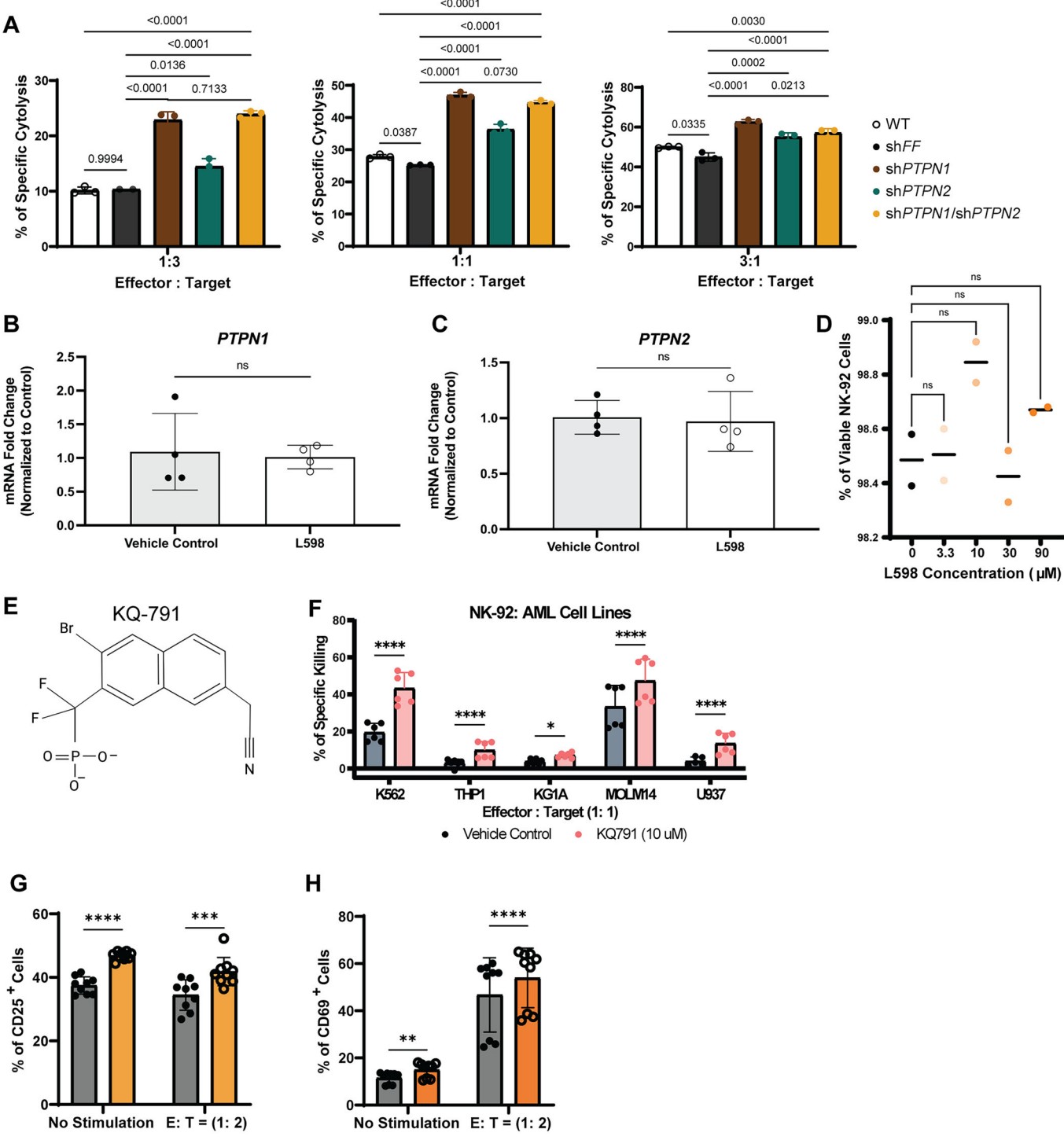

**Figure EV2.   Dual targeting PTPN1 and/or PTPN2 enhances NK cell anti-tumor cytolysis against NK sensitive tumor targets.**

(A) Statistical analysis of flow cytometry-based anti-tumor cytolysis assay (Fig. 2A) using shRNA knockdown NK-92 cells co-cultured with K-562 tumor targets at effector:target (E:T) ratios of 1:3 (left), 1:1 (middle), and 3:1 (right). A representative figure of $n$ = two biological replicates at 1:3 (E:T) ratio and three biological replicates at 1:1 and 3:1 (E:T) ratios, each tested in three technical replicates. Data are presented as mean ± SD. P-values (E:T = 1:3): $p$ = 0.9994 (ns) for WT vs sh*FF*, $p$ = 0.7133 (ns) for sh*PTPN1* vs sh*PTPN1*/sh*PTPN2*, $p$ = 0.0136 (*) for sh*FF* vs sh*PTPN2*, $p$ ≤ 0.0001 (****). P-values (E:T = 1:1): $p$ = 0.0387 (*) for WT vs sh*FF*, $p$ = 0.073 (ns) for sh*PTPN1* vs sh*PTPN1*/sh*PTPN2*, $p$ ≤ 0.0001 (****). P-values (E:T = 3:1): $p$ = 0.0335 (*) for WT vs sh*FF*, $p$ = 0.0213 (*) for sh*PTPN1* vs sh*PTPN1*/sh*PTPN2*, $p$ = 0.003 (**) for WT vs sh*PTPN1*/sh*PTPN2*, $p$ = 0.0002 (***) for sh*FF* vs sh*PTPN2*, $p$ ≤ 0.0001 (****). (One-way ANOVA with Tukey's multiple comparison test). (B, C) qRT-PCR analysis of *PTPN1* (B) and *PTPN2* (C) expression in NK-92 cells treated for 3 days with L598 (30 µM) or vehicle control. Data are presented as mean ± SD. P-values: $p$ = 0.7975 (ns) for *PTPN1*; $p$ = 0.8186 (ns) for *PTPN2* (Unpaired t-test). A representative figure of $n$ = two independent experiments, each tested in four technical replicates is shown. (D) Flow cytometry analysis of NK-92 cell viability following increasing doses of L598 treatment. Results are from two biological replicates; each tested in two or three technical replicates. P-values: $p$ > 0.1234 (ns) (One-way ANOVA with Dunnett's test). (E) A structural representation of KQ791, a dual inhibitor of PTPN1/PTPN2. (F) Flow cytometry-based cytolysis assay of KQ791 (10 µM)-treated or vehicle control-treated NK-92 cells, co-cultured with various acute myeloid leukemia (AML) cell lines. Results are from $n$ = two biological replicates, each tested in three technical replicates. Data are presented as mean ± SD. P-values: $p$ = 0.0127 (*) for KG1A cytolysis; $p$ ≤ 0.0001 (****) (Two-way ANOVA with mixed effects and Šídák's multiple comparisons test). (G, H) Quantifications of the percentage of CD25$^+$ cells and CD69$^+$ NK-92 cells, following treatment with L598 (10 µM) or vehicle control, with or without K562 tumor cell stimulation (E:T = 1:2). Pooled results from $n$ = three biological replicates, each tested in three technical replicates. Data are presented as mean ± SD. P-values (CD25): $p$ = 0.0003 (***) at E:T (1:2), $p$ ≤ 0.0001 (****) at no stimulation. P-values (CD69): $p$ = 0.0025 (**) at no stimulation, $p$ ≤ 0.0001 (****) at E:T (1:2). (Two-way ANOVA with Šídák's multiple comparisons test).

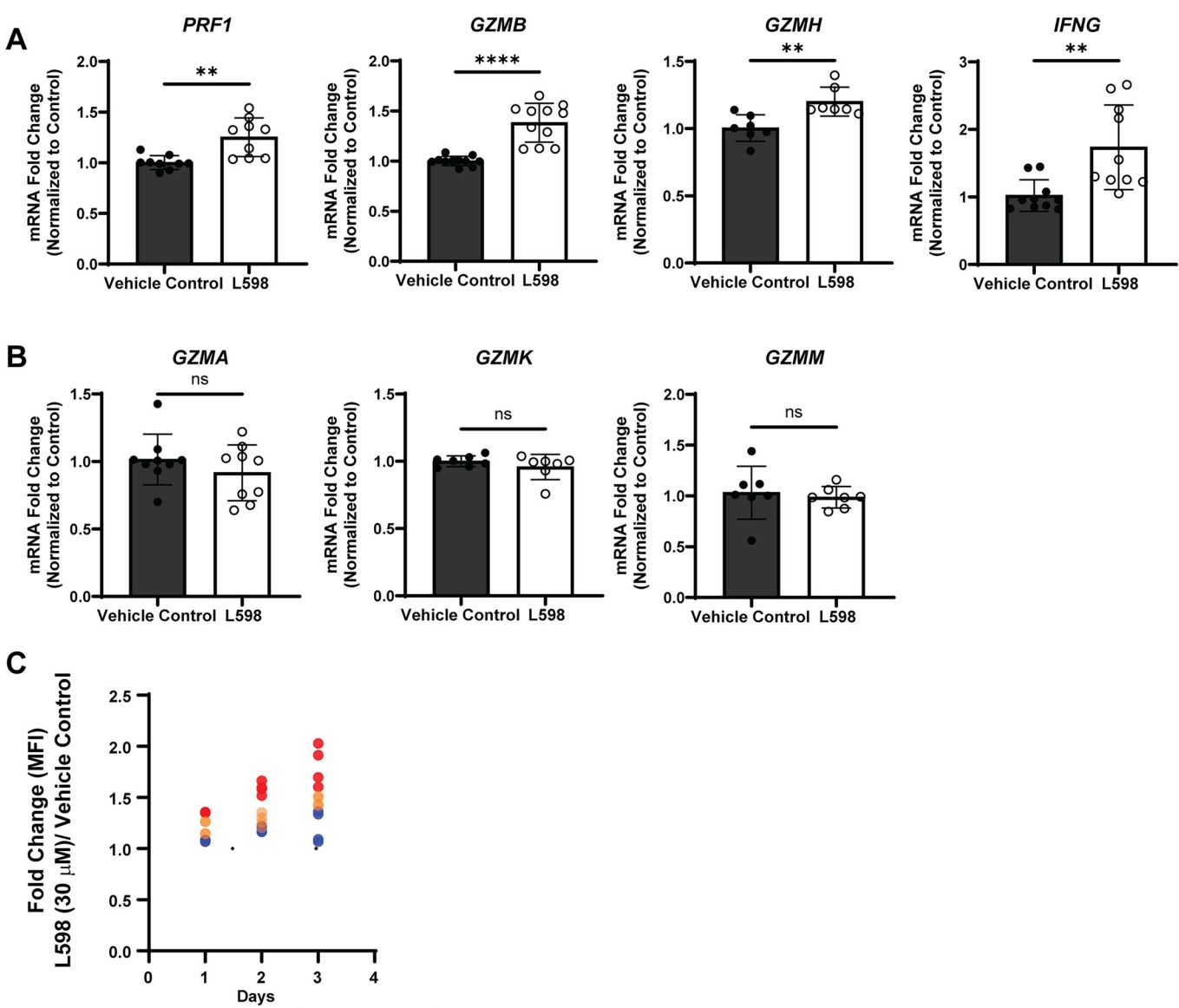

**Figure EV3. Transcriptional expression of human granzymes (A, B, H, K, M), perforin, and interferon-γ in NK-92 cells with dual PTPN1/PTPN2 inhibition.**

(A) mRNA expression of cytolytic effector molecules: perforin (*PRF1*), granzyme B (*GZMB*), granzyme H (*GZMH*) and IFN-γ (*IFNG*) in NK-92 cells after 3-days treatment of L598 (30 μM) or vehicle control. Results are from $n =$ three biological replicates for *PRF1*, *GZMB* and *IFNG*, and from $n =$ two biological replicates for *GZMH*, each tested in three or four technical replicates. Fold change was calculated against vehicle control of the Treatment group. Data are presented as mean ± SD. P-values: $p = 0.004$ (**) for *PRF1* and *GZMH*, $p = 0.0057$ (**) for *IFNG*, $p \leq 0.0001$ (****) for *GZMB* (Unpaired t test with Welch's correction). (B) mRNA expression of granzyme A (*GZMA*), granzyme K (*GZMK*), and granzyme M (*GZMM*) did not significantly change after L598 (30 μM) treatment for 3 days compared with the vehicle control group in NK-92 cells. Results are from $n =$ two biological replicates; each tested in three or four technical replicates. Data are presented as mean ± SD. P-values: $p = 0.3048$ (ns) for *GZMA*, $p = 0.3028$ (ns) for *GZMK*, $p = 0.6761$ (ns) for *GZMM*. (Unpaired t test with Welch's correction). (C) Fold changes in intracellular perforin, granzyme B and granzyme A protein expressions were measured by flow cytometry in NK-92 cells with L598 (30 μM) or vehicle control for 3 days. Results are from $n =$ two biological replicates; each tested in two technical replicates.

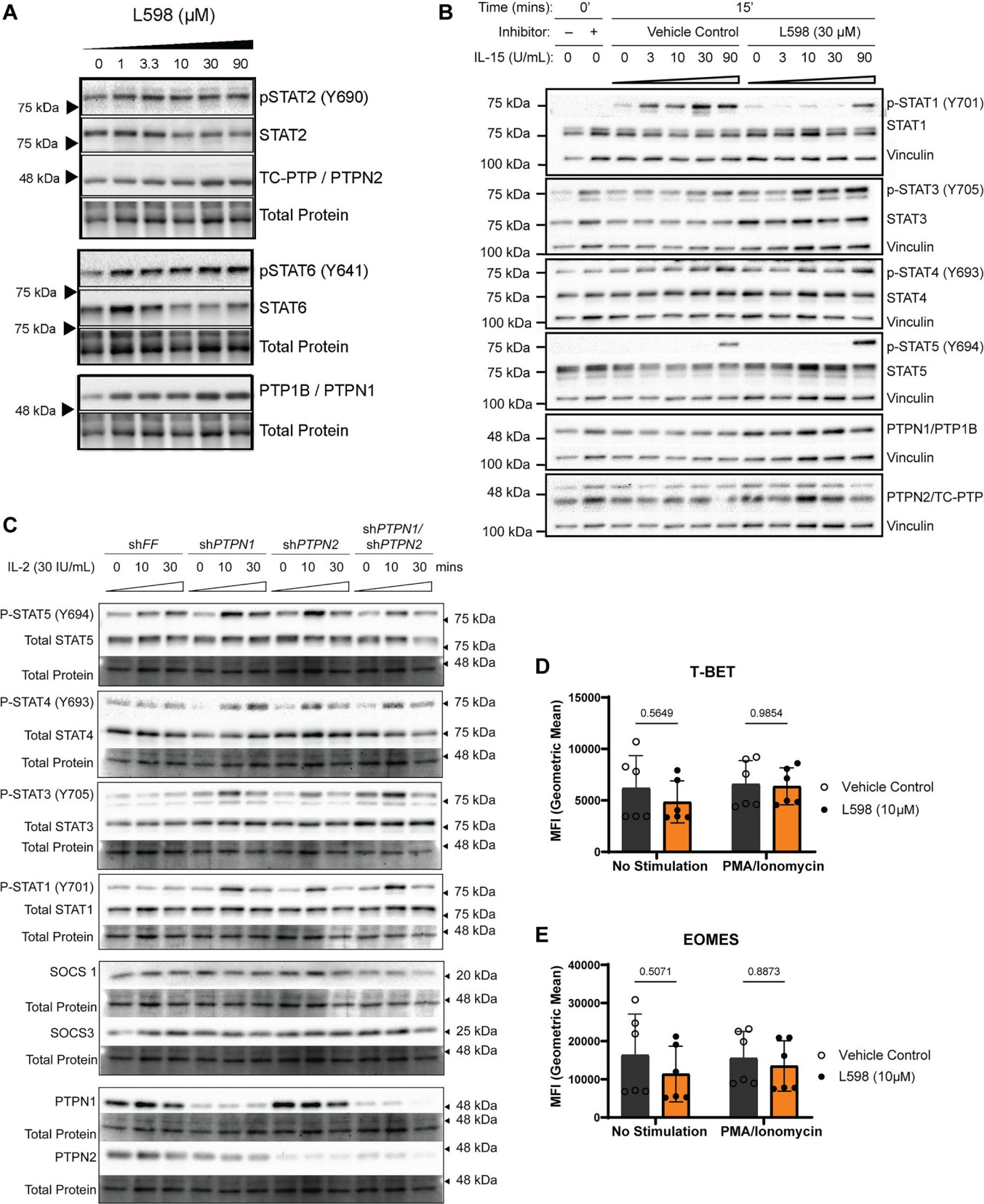

◀ **Figure EV4.    Dual targeting of PTPN1 and PTPN2 modulates STAT signaling downstream of IL-2 or IL-15 in NK-92 cells.**

(A) Western blot analysis of STAT2 (Y690) and STAT6 (Y641) phosphorylation in NK-92 cells treated with L598 (0–90 μM) and cultured in the presence of IL-2 (100 IU/mL) for 3 days. A representative figure from $n$ = two biological replicates is shown. Total protein (~ 48 kDa) is used as the loading control. (p)-STAT6 (> 75 kDa) and Granzyme B (~ 35 kDa) shown in Fig. 2I, were detected on the same membrane; accordingly, the same total protein loading control is presented for both. Likewise, the same membrane was initially probed for (p)-STAT1 (Fig. 4A) and PTPN1 (Fig. EV4A), then stripped and re-probed for (p)-STAT2 and PTPN2 (both shown in Fig. EV4A); therefore, the same loading control is shown for all panels. (B) Western blot analysis of phosphorylated STAT signaling in NK-92 cells treated with dual inhibitor L598 (30 μM) or vehicle control, following 15-minute stimulation with increasing doses of IL-15 (0, 3, 10, 30, 90 IU/mL). A representative figure from $n$ = two biological replicates is shown. (p)-STAT3 (> 75 kDa) and PTPN1 (> 48 kDa) were detected on the same membrane; accordingly, the same vinculin loading control is presented for both. Likewise, the same membrane was blotted for (p)-STAT5 (> 75 kDa) and PTPN2 (< 48 kDa) and the same vinculin loading control is presented for both. (C) Western blot analysis of phosphorylated STAT signaling in shRNA knockdown NK-92 cells stimulated with IL-2 (30 IU/mL) for 10 or 30 min. A representative figure from n = two biological replicates is shown. The same membrane was used to detect (p)-STAT1 (> 75 kDa) and SOCS1 (~ 20 kDa); therefore, the same total protein loading control is shown. Likewise, (p)-STAT4 ( > 75 kDa) and PTPN1 (> 48 kDa) were detected on the same membrane, and (p)-STAT5 (> 75 kDa), PTPN2 (< 48 kDa), and SOCS3 (~ 25 kDa) were detected on the same membrane; therefore, the same total protein loading controls are shown for each group. (D, E) Flow cytometry analysis of transcription factors TBET and EOMES in NK-92 cells treated with L598 (10 μM) or vehicle control, under PMA/ionomycin stimulation or unstimulated conditions. $N$ = two biological replicates, each tested in three technical replicates are shown. Data are presented as mean ± SD. P-values: $p$ = 0.5649 (ns) for no stimulation; $p$ = 0.9854 (ns) for PMA/ionomycin stimulation (Two-way ANOVA with Šídák's test).

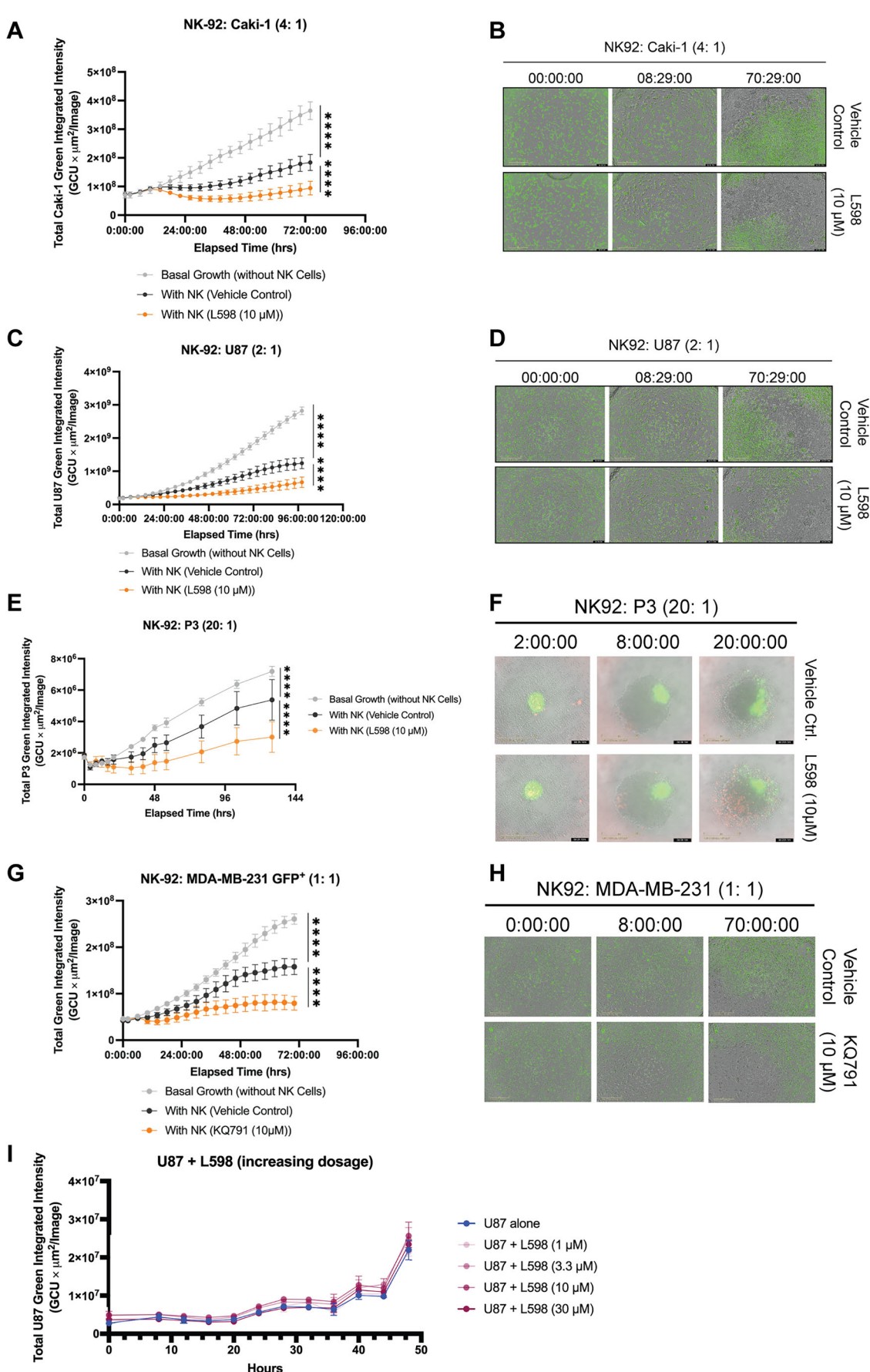

**Figure EV5.   Dual inhibition of PTPN1/PTPN2 enhances NK cell-mediated cytolysis of solid tumor cell lines in vitro.**

(**A–D**) IncuCyte-based real-time cytolysis assay of NK-92 cells treated with L598 (10 μM) or vehicle control, co-cultured with eGFP⁺ Caki-1 kidney cancer cells (E:T = 4:1) in (**A, B**) or eGFP⁺ U87 glioblastoma cells (E:T = 2:1) in (**C, D**) over 3 days. Green fluorescence quantification (tumor cell viability) is shown in (**A**) and (**C**); representative merged images of bright-field and green fluorescence channels are shown in (**B**) and (**D**). Experiments were performed in n = three biological replicates, each tested in six technical replicates. Data are presented as mean ± SD. *P*-values: $p \leq 0.0001$ (****) (Two-way ANOVA with Tukey's test). (**E, F**) IncuCyte-based cytolysis assay of NK-92 cells treated with L598 (10 μM) or vehicle control against eGFP⁺ P3 tumor cells (E:T = 20:1) over 3 days. Quantification of green fluorescence is shown in (**E**). Representative live-cell images combining bright-field, green (eGFP), and red fluorescence (cleaved caspase-3/7 substrate) are shown in (**F**). Experiments were performed in n = three biological replicates, each tested in six technical replicates. Data are presented as mean ± SD. *P*-values: $p \leq 0.0001$ (****) (Two-way ANOVA with Tukey's test). (**G, H**) IncuCyte-based cytolysis assay of NK-92 cells pre-treated with KQ791 (10 μM) or vehicle control, co-cultured with eGFP⁺ MDA-MB-231 cells (E:T = 1:1) for 3 days. Quantification of green fluorescence is shown in (**G**), and representative images of merged bright-field and green fluorescence are shown in (**H**). Experiments were performed in n = three biological replicates, each tested in six technical replicates. Data are presented as mean ± SD. *P*-values: $p \leq 0.0001$ (****) (Two-way ANOVA with Tukey's test). (**I**) Tumor growth assay of eGFP⁺ U87 cells treated with increasing concentrations of L598 (1–30 μM). Experiments were performed in n = three biological replicates; each tested in two technical replicates. Data are presented as mean ± SD.

