## [Peer Review File · EMBO Reports]

PTPN1/PTPN2 inhibition improves NK cancer therapy by enhancing IL-2 and mitigating TGF β 1 responses

Chu-Han Feng, Linda Peltier, Tiffanie Chouleur, Milea DiPonzio, Isabelle Aubry, Alexandre Poirier, Zuzet Martinez Cordova, Yunyun Shen, Sébastien Tabariès, Xiaona Cao, Guojun Chen, Andreas Bikfalvi, Silvia Vidal, Peter Siegel, Pierre Laneuville, and Michel Tremblay

Corresponding author(s): Michel Tremblay (michel.tremblay@mcgill.ca)

Review Timeline:

Submission Date:	7th Aug 25
Editorial Decision:	21st Aug 25
Revision Received:	24th Dec 25
Editorial Decision:	22nd Jan 26
Revision Received:	14th Feb 26
Accepted:	26th Feb 26

Editor: Achim Breiling

Transaction Report: The first round of review of this manuscript was performed at another journal.

Dear Dr. Tremblay

Thank you for the submission of your research manuscript to EMBO reports, accompanied with referee reports from a submission to a journal outside EMBO press, and your detailed point-by-point response. I have now received the report from the advisor/arbitrator that was asked to evaluate your manuscript, which can be found at the end of this email.

As you will see, the advisor thinks that your manuscript is of high interest, that you addressed well the concerns of the referees and that the study should be published in EMBO reports. However, the advisor has a major concern regarding the data presentation and some suggestions to improve the manuscript, I ask you to address in a final revised version. Please also provide a final point-by-point response to these final points.

The manuscript needs also formatting according to our journal style. Therefore, please carefully review the instructions that follow below.

- 1) a .docx formatted version of the final manuscript text (including legends for main figures, EV figures and tables), but without the figures included. Figure legends should be compiled at the end of the manuscript text.
- 2) individual production quality figure files as .eps, .tif, .jpg (one file per figure), of main figures and EV figures. Please upload these as separate, individual files upon re-submission.

The Expanded View format, which will be displayed in the main HTML of the paper in a collapsible format, has replaced the Supplementary information. You can submit up to 6 images as Expanded View. Please follow the nomenclature Figure EV1, Figure EV2 etc. The figure legend for these should be included in the main manuscript document file in a section called Expanded View Figure Legends after the main Figure Legends section. Additional Supplementary material should be supplied as a single pdf file labeled Appendix. The Appendix should have page numbers and needs to include a table of content on the first page (with page numbers) and legends for all content, but not a full title page with author names and affiliations (it is sufficient to state 'Appendix for ...' followed by the title of the paper). Please follow the nomenclature Appendix Figure Sx, Appendix Table Sx etc. throughout the text, and also label the figures and tables according to this nomenclature.

- 3) a .docx formatted letter INCLUDING the reviewers' reports and your final point-by-point response to the points of the arbitrator. As part of the EMBO Press transparent editorial process, the point-by-point response is part of the Review Process File (RPF), which will be published alongside your paper.

- 4) a complete author checklist, which you can download from our author guidelines

(<https://www.embopress.org/page/journal/14693178/authorguide>). Please insert page numbers in the checklist to indicate where the requested information can be found in the manuscript. The completed author checklist will also be part of the RPF.

- 5) that primary datasets produced in this study (e.g. RNA-seq, ChIP-seq, structural and array data) are deposited in an appropriate public database. If no primary datasets have been deposited, please also state this in a dedicated section (e.g. 'No primary datasets have been generated and deposited'), see below.

The accession numbers and database should be listed in a formal "Data Availability" section that follows the model below. This is now mandatory (like the COI statement). Please note that the Data Availability Section is restricted to new primary data that are part of this study. As indicated above, if no primary datasets have been deposited, please state this also.

Data availability

6) We now request the publication of original source data with the aim of making primary data more accessible and transparent to the reader. You will receive a separate email with instructions for providing source data with your revised manuscript, including information how to upload and organize the files.

8) Regarding data quantification and statistics, please make sure that the number "n" for how many independent experiments were performed, their nature (biological versus technical replicates), the bars and error bars (e.g. SEM, SD) and the test used to calculate p-values is indicated in the respective figure legends (also for EV and Appendix figures). Please also check that all the p-values are explained in the legend, and that these fit to those shown in the figure. Please provide statistical testing where applicable. Please avoid the phrase 'independent experiment', but clearly state if these were biological or technical replicates. Please also indicate (e.g. with n.s.) if testing was performed, but the differences are not significant. In case n=2, please show the data as separate datapoints without error bars and statistics. See also: <http://www.embopress.org/page/journal/14693178/authorguide#statisticalanalysis>

Please add to each legend (main, EV figures and Appendix Figures, where applicable) a 'Data Information' section explaining the statistics used or providing information regarding replicates and scales. See:

9) Please add scale bars of similar style and thickness to all microscopic images, using clearly visible black or white bars (depending on the background). Please place these in the lower right corner of the images themselves. Please do not write on or near the bars in the image but define the size in the respective figure legend.

10) Please use our reference format:

12) We now use CRediT to specify the contributions of each author in the journal submission system. CRediT replaces the author contribution section. Please use the free text box to provide more detailed descriptions and do NOT provide your final manuscript text file with an author contributions section. See also our guide to authors: <https://www.embopress.org/page/journal/14693178/authorguide#authorshipguidelines>

13) All Materials and Methods need to be described in the main text using our 'Structured Methods' format, which is required for all research articles. According to this format, the Methods section should include a Reagents and Tools Table (listing key reagents, experimental models, software, and relevant equipment and including their sources and relevant identifiers), uploaded as separate file, and a Methods section in which we encourage the authors to describe their methods using a step-by-step

protocol format with bullet points, to facilitate the adoption of the methodologies across labs. More information on how to adhere to this format as well as downloadable templates (.doc) for the Reagents and Tools Table can be found in our author guidelines (section 'Structured Methods'):

Please move all the information presently provided in the three 'Supplemental Method Tables' to the Reagents and Tools Table and adjust the callouts.

14) Please order the manuscript sections like this, using only these names:

Title page - Abstract - Keywords - Introduction - Results - Discussion - Methods - Data availability section - Acknowledgements (please put here all the funding information) - Disclosure and Competing Interests Statement - References - Figure legends - Expanded View Figure legends

15) Please make sure that all the funding information is also entered into the online submission system and that it is complete and similar to the one in the acknowledgement section of the manuscript text file.

16) Please confirm that for all Western blot panels in the manuscript the loading control was run on the same gel as the other proteins detected. Please note that we discourage comparisons between samples on different gels/blots, even if the samples derive from one experiment, as confounding factors reduce comparability. If unavoidable, the figure legend must state that the samples derive from the same experiment and that gels/blots were processed in parallel. If a 'representative' loading control is shown for multiple gels/blots, the intra-gel controls should be shown in the source data files and the figure legends should describe the data displayed accurately. See our author guidelines:

<https://www.embopress.org/page/journal/14693178/authorguide#datapresentationformat> (section 'Electrophoretic gels and blots').

and

<https://www.embopress.org/image-integrity>

17) We do not allow supplemental methods. Please move all the methods information to the main manuscript text file.

18) Please provide the abstract written in present tense throughout.

In addition, I would need from you uploaded separately:

As part of the EMBO publication's Transparent Editorial Process, EMBO reports publishes online a Review Process File (RPF) to accompany accepted manuscripts. This File will be published in conjunction with your paper and will include the report of the arbitrator, your point-by-point response and all pertinent correspondence relating to the manuscript.

I look forward to seeing the further revised version of your manuscript when it is ready.

Please let me know if you have questions regarding the revision.

Best,

Arbitrator:

The authors provide a novel application of PTPN1/2 dual inhibitors. Although the mechanism of PTPN1/2 inhibition is not novel, previous work focused on systemic application, rather than on ex vivo treatment of NK cells for adoptive therapy. Furthermore, work in PMID 37794185 was mainly focused on T cells, and data on NK cells are derived from murine cells, not human cells as in the present study.

The authors use NK-92 cells, which is a clinically relevant alternative to primary NK cells for NK cell therapy, so the concern of the reviewers about the reliance on NK cell lines is a rather weak one. The data regarding the effect of PTPN1/2 dual inhibition on NK-92 are in most instances very convincing.

The authors appropriately addressed the comments of the reviewers. Some reviewer comments also reveal that some basic principles of the study were not fully appreciated by the reviewers, such as the comment regarding the effect of L598 on tumor cells, which is not a concern for the setup of the study.

The majority of the study is very well performed and data are presented appropriately. However, one major concern about the data presentation of CB NK cell data (and the statistical analysis + conclusions drawn thereof) is to mention and should be corrected:

Figure 7A, B, C and F: technical replicates are shown as individual data points, which distorts the results, especially given that some donors are included in triplicates, some in duplicates and some in no-replicates. Averaging technical replicates and only showing 5 respectively 10 data points, 1 per donor, seems like a more appropriate way of displaying the data.

Correcting the data for this might lead to a different result regarding the CB NK cell data, which should be appropriately adapted in the manuscript.

Another comment regarding the translational value of the strategy: Do the authors know how long the effect of L598 lasts? Are (or can) NK-92 or CB NK cells re-dosed in patients, and if so, can a steady L598 effect be maintained like this? Discussing this crucial translational point would strengthen the manuscript.

Minor comments:

2B: It is unclear what was done in 2B, were cells treated for 3 days before co-culture or did the co-culture last for three days? Could this please be described more clearly in the text and figure legend?

Investigating the response of individual CB NK cell donors towards L598 is very interesting; however, what conclusions are drawn out of the data? Did the authors find any correlation with CB NK cell (+L598) cytotoxicity and the NK cell subsets or inhibitory/ activating marker expression they show in 7H+I? How were the receptors in 7I chosen? Why were NKG2D and DNAM-1 not assessed, whose ligands are highly expressed on K562?

Furthermore, were the factors documented that were identified in the Rezvani paper (PMID 38238616) about "good" and "bad" cord blood units for CAR NK cell generation (nucleated red blood cell count and time to cryopreservation) and if so, is there a link?

I understand the authors will intensify the work in this direction, which will be interesting to follow up on, but even for the present manuscript, some more comments would be useful for the reader.

Finally, the choice of the papers referring to NK cells in the introduction could be discussed. There are more recent publications that could be quoted such as PMID: 38383621.

In summary, the study is well conducted and adds novelty to the field, and I support publication in EMBO reports. The concerns raised here do not require new experimental data, but addressing them would improve the overall quality of the manuscript.

*Response to the arbitrator comments:
On behalf of Dr. ChuHan Feng and all authors.
Corresponding author: Prof. Michel L. Tremblay*

To begin, we are grateful to the Arbitrator for evaluating and providing constructive comments on our manuscript. Below, we answered all queries raised and included additional experiments requested by the arbitrator (Additional Fig. 1).

Arbitrator: The authors provide a novel application of PTPN1/2 dual inhibitors. Although the mechanism of PTPN1/2 inhibition is not novel, previous work focused on systemic application, rather than on ex vivo treatment of NK cells for adoptive therapy. Furthermore, work in PMID 37794185 was mainly focused on T cells, and data on NK cells are derived from murine cells, not human cells as in the present study.

The authors use NK-92 cells, which is a clinically relevant alternative to primary NK cells for NK cell therapy, so the concern of the reviewers about the reliance on NK cell lines is a rather weak one. The data regarding the effect of PTPN1/2 dual inhibition on NK-92 are in most instances very convincing.

The authors appropriately addressed the comments of the reviewers. Some reviewer comments also reveal that some basic principles of the study were not fully appreciated by the reviewers, such as the comment regarding the effect of L598 on tumor cells, which is not a concern for the setup of the study. The majority of the study is very well performed and data are presented appropriately.

Q1- However, one major concern about the data presentation of CB NK cell data (and the statistical analysis + conclusions drawn thereof) is to mention and should be corrected: Figure 7A, B, C and F: technical replicates are shown as individual data points, which distorts the results, especially given that some donors are included in triplicates, some in duplicates and some in no-replicates. Averaging technical replicates and only showing 5 respectively 10 data points, 1 per donor, seems like a more appropriate way of displaying the data.

A1- We have revised the data analysis by averaging technical replicates to generate representative biological data points. The updated figures now show the mean of technical replicates for each donor.

Q2-Correcting the data for this might lead to a different result regarding the CB NK cell data, which should be appropriately adapted in the manuscript.

A2-We have updated the figures according to this suggestion of averaging technical replicates. The results presented in that manner were updated and included in the manuscript section “Translational implications of dual targeting PTPN1 and PTPN2 to improve umbilical cord blood NK therapy. The conclusion of this section does not change when the dataset is presented as the arbitrator suggested.

Q3- Another comment regarding the translational value of the strategy: Do the authors know how long the effect of L598 lasts? Are (or can) NK-92 or CB NK cells re-dosed in patients, and if so, can a steady L598 effect be maintained like this? Discussing this crucial translational point would strengthen the manuscript.

A3-We thank the Arbitrator for commenting on the translational relevance of the L598 treatment

strategy. Indeed, the wash-off and re-dosing effects of L598 on NK-92 cell signaling and anti-tumor cytotoxicity represent an important aspect of our ongoing work. We have already initiated this approach but towards elucidating in our ongoing work, the molecular targets of PTPN1/PTPN2 at the phosphoproteomic and transcriptomic levels. The wash-off and redosing affect a specific set of downstream genes by transcriptomics, which will require much more work to validate.

In brief, upon wash-off of L598, the expression levels of cytotoxic granules: granzyme B and perforin, gradually returned to baseline, comparable to those in the vehicle control group (as shown in Additional Figure 1 below). Consistently, the enhanced anti-tumor cytotoxicity observed in L598-treated NK-92 cells also declined to baseline after three days of wash-off. Notably, reintroduction of L598 for two days restored the enhanced cytotoxic activity, indicating that the effect of L598 is reversible and can be readily re-induced through re-dosing through a temporal change of approximately three days.

Given that persistent STAT3 activation has been associated with oncogenic transformation across various cell types, the transient, controllable effect of L598 may, in fact, be beneficial, as it prevents sustained STAT3 activation in NK cells while allowing periodic re-dosing to maintain therapeutic efficacy. This suggests that, in a clinical setting, NK-92 cells could be re-dosed with L598 at defined intervals to sustain anti-tumor activity without inducing persistent cancerous signalling.

Please note that this figure is included to answer the arbitrator's comment. It is part of the continuing work that we hope to complete in the coming spring. This is not crucial to the present manuscript, as we are pursuing an extensive study to clarify the downstream effects of PTPN1/PTPN2 inhibition with L598.

Figure for referee with unpublished data and its description has been removed upon request by the authors.

Q4 Minor comments: Fig 2B: It is unclear what was done in 2B, were cells treated for 3 days before co-culture or did the co-culture last for three days? Could this please be described more clearly in the text and figure legend?

A4- We clarified the experimental conditions in the legend of Figure 2B. Specifically, NK-92 cells were treated with L598 for one, two, or three days at 30 uM concentration. After treatment, L598 was washed off, and NK-92 cells were co-cultured with K562 tumor target cells for four hours to assess their anti-tumor cytolytic function.

Q5- Investigating the response of individual CB NK cell donors towards L598 is very interesting; however, what conclusions are drawn out of the data? Did the authors find any correlation with CB NK cell (+L598) cytotoxicity and the NK cell subsets or inhibitory/ activating marker expression they show in 7H+I? How were the receptors in 7I chosen? Why were NKG2D and DNAM-1 not assessed, whose ligands are highly expressed on K562?

A5- The observed donor-specific differences in CB NK cell responses to L598 have brought us to a follow-up project aimed at identifying donor-specific markers that could facilitate the development of more precise CB NK cell therapies. We plan to share these findings in a future publication as they are ongoing and outside the scope of this manuscript. These lengthier experiments include analyses of potential correlations between NK cell subsets, inhibitory/activating receptor expression, and anti-tumour cytotoxicity.

-In the present study, we selected KIRs and cytokine receptors for initial immunophenotyping because KIRs are highly polymorphic and PTPN1/PTPN2 regulate cytokine receptor signaling that depends on tyrosine phosphorylation; both of which could contribute to heterogeneity in NK cell cytotoxic responses. We also included natural cytotoxicity receptors (NCRs), which possess immunoglobulin or immunoglobulin-like domains similar to those of KIRs, to explore whether NCR expression might underlie donor variability.

-We appreciate the Arbitrator's suggestion to include NKG2D and DNAM-1, whose ligands are highly expressed on K562 cells. These receptors are being examined in our ongoing studies to further elucidate their roles in L598-mediated NK cell activation and, most importantly, to characterize the heterogeneous genotype/cytolytic phenotype of the donors' NK cells.

Q6- Furthermore, were the factors documented that were identified in the Rezvani paper (PMID 38238616) about "good" and "bad" cord blood units for CAR NK cell generation (nucleated red blood cell count and time to cryopreservation) and if so, is there a link?

A6- In the study by Rezvani et al. (PMID 38238616), optimal cord blood (CB) units for CAR NK cell generation were defined as those with a time from collection to cryopreservation \leq of 24 hours and a nucleated red blood cell (NRBC) count \leq of 8×10^7 . Our cord blood NK cell samples met these criteria because we follow this protocol for isolation and culture. However, despite satisfying these parameters, we still observed variability in baseline cytolytic potency and responsiveness to L598 treatment. We added a sentence in the result section to clarify that the source of CB NK cells was from the optimal cord blood units.

Q7- I understand the authors will intensify the work in this direction, which will be interesting to follow up on, but even for the present manuscript, some more comments would be useful for the reader.

A7- We recognize the importance of adding more detail in the experimental design, results and

Conclusion. We trust that we have indeed clarified and extended the discussion and improved the identification of optimal cord blood units based on their cytolytic activities.

Q8- Finally, the choice of the papers referring to NK cells in the introduction could be discussed. There are more recent publications that could be quoted such as PMID: 38383621.

A8- We thank the Arbitrator for these comments. We agree; accordingly, we incorporated more recent and appropriate references into the manuscript.

Q9- In summary, the study is well conducted and adds novelty to the field, and I support publication in EMBO Reports. The concerns raised here do not require new experimental data, but addressing them would improve the overall quality of the manuscript.

A9- We thank once more the arbitrator for the generous and detailed remarks that were made throughout the manuscript. We hope that we have responded reasonably to all the questions and suggestions.

Submitted on behalf of all authors.

Dr. ChuHan Feng (first author) and Dr. Michel L. Tremblay (corresponding author).

December 22nd, 2025

Dear Dr. Tremblay

Thank you for the submission of your revised manuscript to our editorial offices. I now went through this and your p-b-p-response and consider the points by the arbitrator as adequately addressed.

Before we can proceed with formal acceptance, I have the editorial requests below I ask you to address in a final revised manuscript. Please also provide a final p-b-p-response to the editorial requests.

Editorial requests:

- I would suggest this minimally changed title:

PTPN1/PTPN2 inhibition improves NK cancer therapy by enhancing IL-2 and mitigating TGF β 1 responses

- Please remove the mention of the Appendix from the title page.

- Please use our reference format:

<https://link.springer.com/journal/44319/submission-guidelines#cms-Reference-guidelines>

- There is a file named 'Supplemental Materials' uploaded, that also contains methods information but also figures and tables partly overlapping with the Appendix. None of these supplemental items in called out in the manuscript text file. If this information should be part of the final paper, please add all these items to the final Appendix file, reorganise the items and their callouts. We need one final file with supplemental information but called 'Appendix' (and the figures named 'Appendix Figure Sx'). Finally, please delete the file 'Supplemental Materials'.

- All the methods information should be present in the main manuscript text file. Thus, please add the information presently provided in the 'Supplemental Materials' to the main manuscript text file.

- Please include all antibody and primer information into the Reagents & Tools table and update all related callouts. Then, please remove Appendix Tables S1, S2 and S3 from the Appendix.

- Please check again that the number "n" for how many independent experiments were performed, their nature (biological versus technical replicates), the bars and error bars (e.g. SEM, SD) and the test used to calculate p-values is indicated in the respective figure legends (main, EV and Appendix figures). Please also check that all the p-values are explained in the legend, and that these fit to those shown in the figure. Please provide statistical testing where applicable. Please avoid the phrase 'independent experiment' but clearly state if these were biological or technical replicates. Please also indicate (e.g. with n.s.) if testing was performed, but the differences are not significant. In case n=2, please show the data as separate datapoints without error bars and statistics. See also:

<https://link.springer.com/journal/44319/submission-guidelines#cms-Figure-and-data-presentation>

If n<5, please show single datapoints for diagrams. Moreover:

- Please note that the legend for figure 6 is not provided in the sequential manner. This needs to be rectified.

- Please define the annotated p values ****/***/**/* as well as provide the exact p-values for the same in the legend of figure 2E, 7J, EV5 A, C, E as appropriate.

- Please note that the exact p values are not provided in the legends of figures 1A, J; 2B, C, D, F; 4C, D, E, F; 5C, 6B, D, F; EV1 A, B, D, F, G-J; EV2 A, F-H; EV3 A, B; S3 A, B

- Please indicate the statistical test used for data analysis in the legends of figures 2E, 7J, EV5 A, C, E

- Please note that information related to n is missing in the legends of figures EV5 A, C, E.

- Please note that the error bars are not defined in the legends of figures 2C, D; 3E, 5E, 6B, D, F; EV2 A, B, C, F, G, H; EV3 A; EV4 D, E; EV5 A, C, E; S3 A, B

- Please make sure that all the funding information is also entered into the online submission system and that it is complete and similar to the one in the acknowledgement section of the manuscript text file. Presently, the grant from European Joint Program for Rare Diseases (EJPRD2019-40) is only mentioned in the Acknowledgements and a grant from Genome Canada is listed only listed in the submission system. Please check.

- Please make sure that each figure panel is called out separately and that the panels are called out sequentially. Presently, it seems there is no callout for Appendix Figure S6. Please check.

- During our regular image integrity checks we noted that several Western blot images have been re-used. We noted a

Possible reuse between Figure 2I and Figure EV4A - total protein

Possible reuse between Figure 2I and Sup Figure 5A - total protein
Possible reuse within Figure 4B - Vinculin
Possible reuse between Figure 4A and Figure EV4A
Possible reuse between Figure 4A and Sup Figure 5A
Possible reuse between Figure 4B and App Figure S2A&B Vinculin
Possible reuse between Figure EV2E and App Supp Figure 2D
Possible reuse within Figure EV4 A & B & C
Possible reuse between Figure EV4A and App Figure S5A
Possible reuse between Figure EV4B & C and Sup Figure S6 A & B
Possible reuse between Page EV5B and Sup Figure 8B
Possible reuse within Figure Supp 5A
Possible reuse between Sup Figure 5 B&C and Appendix Fig S2 A & B
Possible reuse within Figure research article Sup Figure 6 A and B
Possible reuse between Sup Figure 6C & Appendix Fig S2C
Possible reuse between Supp Figure 7 C and Appendix Fig S3C - Same figure.
Possible reuse between Sup Figure 10 A and Appendix Fig S4 A - Same figure
Possible reuse between Sup Figure 11 A&B and Appendix Fig S5 A & B

I think these images are re-used intentionally, as proteins were run on the same gel. Nevertheless, to increase clarity, could you please always indicate in the respective figure legends when the same image is used for different panels and explain the re-use. I think most of these duplications are also caused by the overlapping panels in the Appendix and the file 'Supplemental Materials'. This will be fine if the panel is shown only once in the final submission. Please check.

In addition, I would need from you uploaded separately:

- a schematic summary figure as separate file that provides a sketch of the major findings (not a data image) in jpeg or tiff format (with the exact width of 550 pixels and a height of not more than 400 pixels) that can be used as a visual synopsis on our website. The image provided contains partly text that is not readable. Please provide this with bigger fonts.

I look forward to seeing the further revised version of your manuscript when it is ready. Please let me know if you have questions regarding the revision.

Best,

Editorial requests:

- I would suggest this minimally changed title:

PTPN1/PTPN2 inhibition improves NK cancer therapy by enhancing IL-2 and mitigating TGF β 1 responses

- Response: We have corrected the title.

- Please remove the mention of the Appendix from the title page.

- Response: We have removed the Appendix from the title page.

- Please use our reference format:

<https://link.springer.com/journal/44319/submission-guidelines#cms-Reference-guidelines>

- Response: We used the EMBO Reports citation format downloaded from the following website: <https://paperpile.com/s/embo-reports-citation-style/>.

- There is a file named 'Supplemental Materials' uploaded, that also contains methods information but also figures and tables partly overlapping with the Appendix. None of these supplemental items in called out in the manuscript text file. If this information should be part of the final paper, please add all these items to the final Appendix file, reorganise the items and their callouts. We need one final file with supplemental information but called 'Appendix' (and the figures named 'Appendix Figure Sx'). Finally, please delete the file 'Supplemental Materials'.

- Response: We have removed the Supplemental Materials. They were submitted during our initial submission and should not be included in the formal final submission. The information in the Appendix file is correct, and the Supplemental Materials file is deleted in this final submission.

- All the methods information should be present in the main manuscript text file. Thus, please add the information presently provided in the 'Supplemental Materials' to the main manuscript text file.

- Response: Yes, all the methods information is present in the main manuscript text file. The final manuscript text file contains methods provided previously in ‘Supplemental Materials’

- Please include all antibody and primer information into the Reagents & Tools table and update all related callouts. Then, please remove Appendix Tables S1, S2 and S3 from the Appendix.

- Response: Appendix Tables S1, S2 and S3 have been removed and information has been transferred into the Reagents & Tools table.

- Please check again that the number "n" for how many independent experiments were performed, their nature (biological versus technical replicates), the bars and error bars (e.g. SEM, SD) and the test used to calculate p-values is indicated in the respective figure legends (main, EV and Appendix figures). Please also check that all the p-values are explained in the legend, and that these fit to those shown in the figure. Please provide statistical testing where applicable. Please avoid the phrase 'independent experiment' but clearly state if these were biological or technical replicates. Please also indicate (e.g. with n.s.) if testing was performed, but the differences are not significant. In case n=2, please show the data as separate datapoints without error bars and statistics. See also:

<https://link.springer.com/journal/44319/submission-guidelines#cms-Figure-and-data-presentation>

- Response: We have specified the “n” number’s nature: biological versus technical replicates, in the figure legends. We have indicated p-values and the error bars in the figure legends.

- Response: We have modified the figures to show single datapoints for diagrams when n < 5.

Moreover:

- Please note that the legend for figure 6 is not provided in the sequential manner. This needs to be rectified.

- Response: We have corrected the order in the legend in figure 6.

- Please define the annotated p values ****/****/**/* as well as provide the exact p-values for the same in the legend of figure 2E, 7J, EV5 A, C, E as appropriate.

- Response: We have specified p-values and corrected legend for figures 2E, 7J, EV5A, C, E.

- Please indicate the statistical test used for data analysis in the legends of figures 2E, 7J, EV5 A, C, E

- Response: We have included statistical tests for figure 2E, 7J, EV5A, C, E.

- Please note that information related to n is missing in the legends of figures EV5 A, C, E.

- Response: We have included n number information for figures EV5A, C, E.

- Please note that the exact p values are not provided in the legends of figures 1A, J; 2B, C, D, F; 4C, D, E, F; 5C, 6B, D, F; EV1 A, B, D, F, G-J; EV2 A, F-H; EV3 A, B; S3 A, B

Response:

- We have provided p-values for 1A, 1J, 2B, 2C, 2D, 2F, 4C, 4D, 4E, 4F, 5C, 6B, 6D, 6F, EV1A, EV1B, EV2F, EV2G, EV2H, EV3A, EV3B, S3A, S3B in legends in our previous submission.
- We have provided p-values for EV1D, EV1F, EV1G – J, EV2A in figures and with legend referring to the figure and have included p-values in the figure legends in the corrected version.
- We have included p-values in the figure for 2C and 2D and clarified p-values for 1A, 1J, 2B, 2F, 4C, 4D, 4E, 4F, 5C, 6B, 6D, 6F, EV1A, EV1B, EV1D, EV1F, EV1G-J, EV2A, EV2F, EV2G, EV2H, EV3A, EV3B, S3A, S3B.

- Please note that the error bars are not defined in the legends of figures 2C, D; 3E, 5E, 6B, D, F; EV2 A, B, C, F, G, H; EV3 A; EV4 D, E; EV5 A, C, E; S3 A, B

- Response: We have defined the error bars in figures 2C, D; 3E, 5E, 6B, D, F; EV2 A, B, C, F, G, H; EV3 A; EV4 D, E; EV5 A, C, E; S3 A, B

- Please make sure that all the funding information is also entered into the online submission system and that it is complete and similar to the one in the acknowledgement section of the manuscript text file. Presently, the grant from European Joint Program for Rare Diseases (EJPRD2019-40) is only mentioned in the Acknowledgements and a grant from Genome Canada is listed only listed in the submission system. Please check.

- Response: We have included funding information about the Genome Canada /Genome Quebec GAPP awards in the Acknowledgements and added the EJPRD2019-40 in this final submission.

- Please make sure that each figure panel is called out separately and that the panels are called out sequentially. Presently, it seems there is no callout for Appendix Figure S6. Please check.

- Response: We have added a callout for Appendix Figure S6 in the final manuscript.

- During our regular image integrity checks we noted that several Western blot images have been re-used. We noted a

- We have added explanations in corresponding figure legends. Please see the following responses:

1. Possible reuse between Figure 2I and Figure EV4A - total protein

- Explained in Figure EV4A that the same total protein was used for Figure 2I granzyme B and Figure EV4A STAT6.

2. Possible reuse between Figure 2I and Sup Figure 5A - total protein

- Sup Figure 5A was not shown in the final version of the manuscript.

3. Possible reuse within Figure 4B - Vinculin

- Explained in Figure legend 4B: The same membrane was sequentially probed for (p)-STAT3 and (p)-STAT5; therefore, the same loading control was shown for both.

4. Possible reuse between Figure 4A and Figure EV4A

- Explained in Figure legend EV4A: the same membrane was initially probed for (p)-STAT1 (Figure 4A) and PTPN1 (Figure EV4A), then stripped and re-probed for (p)-STAT2 and PTPN2 (both shown in Figure EV4A); therefore, the same loading control was shown for all panels.

5. Possible reuse between Figure 4A and Sup Figure 5A

- Sup Figure 5A was not shown in the final version of the manuscript.

6. Possible reuse between Figure 4B and App Figure S2A&B Vinculin

- Explained in the legend of Figure S2A&2B: The same membrane was probed for (p)-STAT5 (>75 kDa) (Figure 4B) and (p)-AKT (~63 kDa) (Appendix Figure S2A); therefore, the same total protein loading control was shown in (A). Likewise, the same

membrane was probed for (p)-STAT4 (>75 kDa) (Figure 4B) and (p)-p38 MAPK (~35 kDa) (Appendix Figure S2B); therefore, the same total protein loading control was shown in (B).

7. Possible reuse between Figure EV2E and App Supp Figure 2D

- App Supp Figure 2D were not shown in the final version of the manuscript.

8. Possible reuse within Figure EV4 A & B & C

- Explained in the figure legend of Figure EV4A, EV4B and EV4C. For EV4B: (p)-STAT3 (>75 kDa) and PTPN1 (>48 kDa) were detected on the same membrane; accordingly, the same vinculin loading control is presented for both. Likewise, the same membrane was blotted for (p)-STAT5 (>75 kDa) and PTPN2 (<48 kDa) and the same vinculin loading control is presented for both.

9. Possible reuse between Figure EV4A and App Figure S5A

- App Figure S5A was not shown in the final version of the manuscript.

10. Possible reuse between Figure EV4B & C and Sup Figure S6 A & B

- Sup Figure S6A & B were not shown in the final version of the final version of the manuscript.

11. Possible reuse between Figure EV5B and Sup Figure 8B

- Sup Figure 8B was not shown in the final version of the final version of the manuscript.

12. Possible reuse within Figure Supp 5A

- Figure Supp 5A was not shown in the final version of the manuscript.

13. Possible reuse between Sup Figure 5 B&C and Appendix Fig S2 A & B

- Sup Figure 5 B&C were not shown in the final version of the manuscript.

14. Possible reuse within Figure research article Sup Figure 6 A and B

- Sup Figure 6A was not shown in the final version of the manuscript.

15. Possible reuse between Sup Figure 6C & Appendix Fig S2C

- Sup Figure 6C was not shown in the final version of the manuscript.

16. Possible reuse between Supp Figure 7 C and Appendix Fig S3C - Same figure.

- Supp Figure 7C was not shown in the final version of the manuscript.

17. Possible reuse between Sup Figure 10 A and Appendix Fig S4 A - Same figure

- Sup Figure 10A was not shown in the final version of the manuscript.

18. Possible reuse between Sup Figure 11 A&B and Appendix Fig S5 A & B

- Sup Figure 11 A&B were not shown in the final version of the final version of the manuscript.

- From Editor: I think these images are re-used intentionally, as proteins were run on the same gel. Nevertheless, to increase clarity, could you please always indicate in the respective figure legends when the same image is used for different panels and explain the re-use. I think most of these duplications are also caused by the overlapping panels in the Appendix and the file 'Supplemental Materials'. This will be fine if the panel is shown only once in the final submission. Please check.

- From Editor: In addition, I would need from you uploaded separately:

- a schematic summary figure as separate file that provides a sketch of the major findings (not a data image) in jpeg or tiff format (with the exact width of 550 pixels and a height of not more than 400 pixels) that can be used as a visual synopsis on our website. The image provided contains partly text that is not readable. Please provide this with bigger fonts.

- Response: we have modified the fonts and provided bigger fonts text.

- From Editor: I look forward to seeing the further revised version of your manuscript when it is ready. Please let me know if you have questions regarding the revision.

Dr. Michel Tremblay
McGill University
Rosalind and Morris Goodman Cancer Centre
1160 Pine Avenue West
Montreal, Quebec H3A 1A3
Canada

Dear Dr. Tremblay,

I am very pleased to accept your manuscript for publication in the next available issue of EMBO reports. Thank you for your contribution to our journal.

You may qualify for financial assistance for your publication charges - either via a Springer Nature fully open access agreement or an EMBO initiative. Check your eligibility: <https://link.springer.com/journal/44319/how-to-publish-with-us>

Yours sincerely,

>>> Please note that it is EMBO Reports policy for the transcript of the editorial process (containing referee reports and your response letter) to be published as an online supplement to each paper. If you do NOT want this, you will need to inform the Editorial Office via email immediately. More information is available here: <https://link.springer.com/partners/embo-press/editorial-policies#Peer%20review>